# Data Distillation for extrapolative protein design through exact preference optimization

**Mostafa Karimi, Sharmi Banerjee**
Amazon
{mkarimii,sharmiba}@amazon.com

**Tommi Jaakkola**
Massachusetts Institute of Technology
tommi@csail.mit.edu

**Bella Dubrov, Shang Shang, Ron Benson**
Amazon
{belladub,shashang,ronben}@amazon.com

## Abstract

The goal of protein design typically involves increasing fitness (extrapolating) beyond what is seen during training (e.g., towards higher stability, stronger binding affinity, etc.). State-of-the-art methods assume that one can safely steer proteins towards such extrapolated regions by learning from pairs alone. We hypothesize that noisy training pairs are not sufficiently informative to capture the fitness gradient and that models learned from pairs specifically may fail to capture three-way relations important for search, e.g., how two alternatives fair relative to a seed. Building on the success of preference alignment models in large language models, we introduce a progressive search method for extrapolative protein design by directly distilling into the model relevant triplet relations. We evaluated our model's performance in designing AAV and GFP proteins and demonstrated that the proposed framework significantly improves effectiveness in extrapolation tasks.

## 1 Introduction

We focus on the challenging task of extrapolative protein design (Chan et al., 2021; Padmakumar et al., 2023; Lee et al., 2023). The problem involves creating novel protein sequences with fitness values above and beyond those seen during training. Extrapolation is challenging for machine learning methods as methods are primarily trained to recognize patterns within the range of the training data (Xu et al., 2020). Existing extrapolative protein design methods can be roughly categorized into two groups. Scorer-based approaches optimize sequences by following a learned score model with the help of reinforcement learning (Lee et al., 2024) or Gibbs sampling (Kirjner et al., 2024), progressively modifying sequences towards higher predicted fitness values. Since the score model in these methods has been trained primarily on sequences belonging to the training region of fitness values, its ability to generalize to sequences further away may be limited. Edit-based methods, on the other hand, effectively learn gradient directions from pairs of examples derived from data or from an estimated score model, and learn to propose sequence improvements. If the learned patters of improvement generalize, then the methods are able to continue to improve the sequences beyond those in the original training region. Current generative models for sequence improvement approximate the gradient direction through differences between protein pairs, e.g., learning the ranking through contrastive discriminatory objective (Chan et al., 2021), token-level machine translation (Padmakumar et al., 2023) and Bradley-Terry (BT) model (Bradley & Terry, 1952) with maximum likelihood objective (Lee et al., 2023). The primary limitation is that noisy pairs alone may not provide sufficient gradient information to effectively guide protein generation.

In this paper, we build on edit-based methods to address these limitations and further enhance the methods with *data distillation*. We draw inspiration from recent successes of preference learning (Christiano et al., 2017; Rafailov et al., 2023) in text generation. In order to improve the extrapolation power of the generative models, we distill carefully selected higher order (triplet) relationships into the models. These triplet relations are meant to better approximate the gradient direction in sequence space and therefore aid in progressive search for better sequences. We focus on improvements

relative to a seed with the help of offline preference learning approaches such as direct preference optimization (DPO) (Rafailov et al., 2023) and efficient exact optimization (EXO) (Ji et al., 2024). We introduce hard triplet relations as additional guidance to the generative model pre-trained from pairs alone. Specifically, given a conditional generative model $P(.|\mathbf{x})$, we identify examples $\mathbf{x}_1 < \mathbf{x}_2 < \mathbf{x}_3$ (based on their fitness) where the current model fails, i.e., $P(\mathbf{x}_1|\mathbf{x}_2) > P(\mathbf{x}_3|\mathbf{x}_2)$. We consider these mistakes similar to harmful text generation in LMs. The proposed *data distillation* approach guides language models towards higher fitness in the extrapolation region while preventing the generation of lower fitness sequences.

We evaluate our method on the well studied Green Fluorescent Proteins (GFP) by Sarkisyan et al. (2016) and Adeno-Associated Virus (AAV) by Bryant et al. (2021). We utilize the carefully created *medium* and *hard* difficulty splits provided by Kirjner et al. (2024). Our contributions are summarized as follows:

- We develop a novel data distillation approach for extrapolative protein design through preference learning. Our approach distills model-dependent hard triplewise ranking into generative model through reverse KL offline preference learning (Section 3.4 for preference data creation and Section 3.5 for general model alignment).

- Our benchmark shows that our approach can drastically improve the performance upon prior methods (Section 5).

- Through ablation studies, we show the importance of training on hard triplewise rankings in comparison to other methods for preference dataset creation (Section 6.1).

- We benchmark our proposed approach against state-of-the-art scorer-based approaches and our approach becomes state-of-the-art in 3 out of 4 datasets (Section 6.2).

- We benchmark the reverse KL preference learning approach against other state-of-the-art preference learning approaches such as DPO (Rafailov et al., 2023), IPO (Azar et al., 2023), IRPO (Pang et al., 2024), NCA (Chen et al., 2024), SPPO (Wu et al., 2024), AOT (Melnyk et al., 2024) (Section 6.3).

## 2 RELATED WORK

**Controllable Biological Sequence Design** The approaches for controlled biological sequence design can be categorized into (i) conditional protein design with control codes such as protein families (Karimi et al., 2020; Madani et al., 2023) (ii) controllable generation via scorer functions such as Gibbs sampling on smoothed fitness landscape (Kirjner et al., 2024) (iii) protein optimization in latent space such as reinforcement learning in latent space of large language models (Lee et al., 2024), property-guided variational auto-encoder based models (Gómez-Bombarelli et al., 2018; Ghaffari et al., 2024) (iv) extrapolative protein design: extrapolating to fitness beyond training data (Chan et al., 2021; Padmakumar et al., 2023; Lee et al., 2023) All extrapolative protein design models have the inherent assumption that extrapolation can be learned sufficiently well through pairwise ranking of protein finesses. Chan et al. (2021) developed a contrastive learning approach of ranking pairs through a discriminator of the latent space and extrapolating biological sequences through traversing it. Padmakumar et al. (2023) proposed to learn local editor for translating sequences with low fitness to sequences with slightly higher fitness through machine translation. Recently, Lee et al. (2023) modeled the ranked pairs through Bradley-Terry (BT) model via a maximum likelihood objective, named align-plm. The align-plm model is an unconditional auto-regressive generative model, so it does not have any notion of seed (starting sequence). Therefore, the align-plm model has been used as an oracle rather than a generative model to rank the fitness of heuristically generated sequences. Particularly, starting from seed sequences, they will exhaustively search all single site mutations and rank them based on align-plm model to choose the top ranking sequences and iteratively continue this process.

**Preference Learning** Aligning language models toward human feedback has improved their capabilities in following instruction (Ouyang et al., 2022) and translation (Kreutzer et al., 2018). LLM alignment originated from the seminal work of Christiano et al. (2017) on reinforcement learning with human feedback (RLHF). However, training RLHF was shown to be challenging due to training instabilities, reward hacking and catastrophic forgetting (Peng et al., 2023). Recently, there has been

a momentum for closed-form and direct optimization of offline preferences. Direct optimization of human feedback can be categorized into sequence likelihood calibration (Zhao et al., 2023), direct preference optimization (DPO) through Bradley-Terry (BT) model (Rafailov et al., 2023; Mitchell, 2023), a more generalized version of DPO named $\Psi$ preference optimization (Azar et al., 2023). It has recently been shown that, under *probability matching* perspective, DPO is optimizing the forward KL divergence $\mathrm{KL}(\pi_\beta^*||\pi_\theta)$ where $\pi_\beta^*$ is the optimal target distribution and $\pi_\theta$ the parameterized model distribution. However, the RLHF objective function is equivalent of minimizing the reverse KL divergence $\mathrm{KL}(\pi_\theta||\pi_\beta^*)$. Recently, efficient exact optimization (EXO) has been proposed to minimize the reverse KL divergence of the general alignment objective (Ji et al., 2024). It has been shown that DPO only covers the support of $\pi_\beta^*$ (mean seeking) and EXO is capturing its modes (mode seeking). Direct preference models not only perform on par with RLHF but are also simpler to implement, single-stage training and computationally efficient in practice.

## 3 METHODS

### 3.1 PROBLEM DEFINITION

Let's assume there is a supervised dataset $\mathrm{D} = \{(\mathbf{x}^n, y^n)\}_{n=1}^N$ with $N$ samples where $\mathbf{x}^n = (x_1^n, \cdots, x_L^n)$ is $n$th protein sequence with length $L$ and $y^n$ is its corresponding fitness value (i.e. stability, binding affinity). Let's assume the fitness value $y$ in dataset D is bounded $y \in [y_{\min}, y_{\max}]$. We define this region as *training region* and try to generate sequences with fitness value $y_{\mathrm{gen}} > y_{\max}$ or $y_{\mathrm{gen}} < y_{\min}$ which is defined as the *extrapolation region*.

### 3.2 OVERVIEW

The core concept behind the proposed method is to gradually learn the higher order relationships among ranked proteins. We start with an auto-regressive unconditional pLM such as Prot-T5-XL (Elnaggar et al., 2021) that is trained on unsupervised data to model $\mathbf{x} \sim P_\theta(.)$ where $\mathbf{x}$ is a generated protein sequence. Inspired by the ICE model (Padmakumar et al., 2023), we trained a local editor with the desired direction (e.g. increasing the binding affinity) to learn the first order relationship among ranked proteins (approximating desired gradient direction through pairs). The model learns to generate $\mathbf{x}_2 \sim P_\theta(.|\mathbf{x}_1)$ where the fitness of $\mathbf{x}_2$ (designed sequence) is expected to be better than $\mathbf{x}_1$ (starting sequence). Inspired by direct preference optimization (DPO) (Rafailov et al., 2023) and EXO (Ji et al., 2024), we aligned the pairwise model based on hard triplets by directly optimizing on newly created preferences where the pairwise model makes the worst mistakes. With this alignment, the model updates its belief of gradient direction from triplewise relationships where the pairwise model is going in the wrong direction. The overall schematic of the proposed method is illustrated in Figure 1.

### 3.3 LOCAL EDITING THROUGH PAIRS

Given a supervised dataset D, we trained a scorer function $f_s$ to predict the fitness of a query sequence. We expect $f_s$ to perform well on the training region and perform poorly on the extrapolation region, since it has not seen these fitness values during its training. Then, following Padmakumar et al. (2023) we generated perturbed sequences by masking-infilling starting from the training sequences (seeds). Scorer function $f_s$ is utilized to assess whether the newly generated pair (seed, sequence) has a small but meaningful improvement in the desired direction. We created the dataset $\mathrm{D}_{\mathrm{pair}} = \{(\mathbf{x}^m, \mathbf{z}^m)\}_{m=1}^M$ with $M$ samples where $f_s(\mathbf{x}^m) < f_s(\mathbf{z}^m)$ if increasing fitness is desired and vice versa. Finally, we fine-tuned the Prot-T5-XL model (Elnaggar et al., 2021) through MLE in an auto-regressive manner to predict the next amino acid: $\mathrm{P}_{\mathrm{pair}}(\mathbf{z}|\mathbf{x}) = \prod_{i=1}^L \mathrm{P}(z_i|\mathbf{z}_{<i}, \mathbf{x})$.

### 3.4 PREFERENCE DATASET CREATION

To better approximate the gradient direction toward improved fitness in the extrapolation region and directly model higher order relationship among proteins, we created a preference dataset of size $K$ based on triplets $\mathrm{D}_{\mathrm{triplet}} = \{(\mathbf{x}_{\mathrm{prompt}}^k, \mathbf{x}_w^k, \mathbf{x}_l^k)\}_{k=1}^K$ where $\mathbf{x}_{\mathrm{prompt}}$ is the seed sequence, $\mathbf{x}_w$ is the *desired* response and $\mathbf{x}_l$ is the *undesired* response. We are interested in increasing fitness by moving

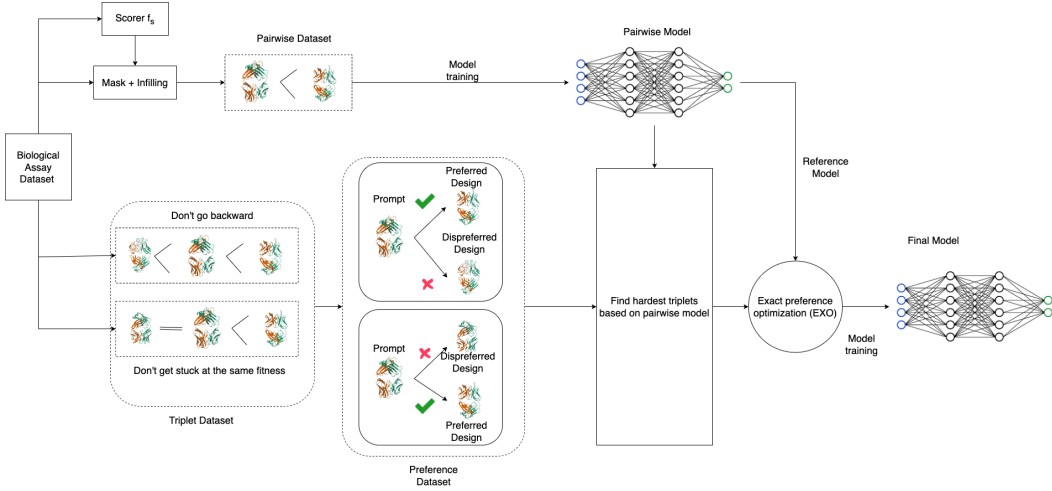

Figure 1: Schematic overview of extrapolative protein design through triplet preference learning.

from $\mathbf{x}_{\text{prompt}}$ toward $\mathbf{x}_w$ where $f_s(\mathbf{x}_{\text{prompt}}) < f_s(\mathbf{x}_w)$ while guarding it against sequences with same or worse fitnesses (undesired ones). Therefore, we create the following preference datasets (i) *Don't go backward*: triplets should satisfy the following order $f_s(\mathbf{x}_l) < f_s(\mathbf{x}_{\text{prompt}}) < f_s(\mathbf{x}_w)$ (ii) *Don't get stuck at the same fitness*: triplets should satisfy the following order $f_s(\mathbf{x}_l) \approx f_s(\mathbf{x}_{\text{prompt}}) < f_s(\mathbf{x}_w)$.

In addition for better approximation of gradient direction and higher order modeling, we would like to focus more on the triplets which are the most confusing for the pairwise model. We mathematically define *hardness* for a given triplet $(\mathbf{x}_{\text{prompt}}, \mathbf{x}_w, \mathbf{x}_l)$ with respect to the pairwise model:

$$S_{\text{hardness}} = \log \mathrm{P}_{\text{pair}}(\mathbf{x}_l|\mathbf{x}_{\text{prompt}}) - \log \mathrm{P}_{\text{pair}}(\mathbf{x}_w|\mathbf{x}_{\text{prompt}}) \tag{1}$$

By definition, triplets with $S_{\text{hardness}} > 0$ are considered to be hard examples, since the pairwise model prefers the undesired sequences to the desired sequences given the prompts, and $S_{\text{hardness}} <= 0$ are easy examples similarly. In our internal experiments, we found out that 65%-70% of the proposed triplets are already easy. We have hypothesized that one should focus on the **hard triplet examples** in the **entire training** fitness range. We have performed ablation studies to investigate the impact of these components. The details of preference creation has been explained in section 4.3.

## 3.5 PREFERENCE LEARNING THROUGH TRIPLETS

We model the triplewise relationship through offline preference optimization recently developed for model alignment. Ji et al. (2024) proposed a general model alignment which is explained in details in supplementary section A. We will utilize their approximate formulation for offline preference setting. Firstly, one can define the empirical distribution $f_\theta$ for a given prompt $\mathbf{x}_{\text{prompt}}$ and a response $\mathbf{x}_j$ where $j \in \{w, l\}$ as follow:

$$\mathrm{P}_{f_\theta}(j|\mathbf{x}_w, \mathbf{x}_l, \mathbf{x}_{\text{prompt}}) = \frac{e^{\beta_\pi \log \frac{\mathrm{P}_\theta(\mathbf{x}_j|\mathbf{x}_{\text{prompt}})}{\mathrm{P}_{\text{pair}}(\mathbf{x}_j|\mathbf{x}_{\text{prompt}})}}}{e^{\beta_\pi \log \frac{\mathrm{P}_\theta(\mathbf{x}_w|\mathbf{x}_{\text{prompt}})}{\mathrm{P}_{\text{pair}}(\mathbf{x}_w|\mathbf{x}_{\text{prompt}})}} + e^{\beta_\pi \log \frac{\mathrm{P}_\theta(\mathbf{x}_l|\mathbf{x}_{\text{prompt}})}{\mathrm{P}_{\text{pair}}(\mathbf{x}_l|\mathbf{x}_{\text{prompt}})}}} \tag{2}$$

Secondly, one can define the empirical distribution $\mathrm{P}_{r_\phi}$ as follow:

$$\mathrm{P}_{r_\phi}(j|\mathbf{x}_w, \mathbf{x}_l, \mathbf{x}_{\text{prompt}}) = \begin{cases} 1 - \epsilon : & j = w \\ \epsilon : & j = l \end{cases}$$

Where $\epsilon$ is a small number (e.g. $10^{-5}$). Finally, the offline alignment objective is the reverse KL between $\mathrm{P}_{f_\theta}$ and $\mathrm{P}_{r_\phi}$:

$$\mathcal{L}_{\mathrm{exo}} = \mathbb{E}_{\{\mathbf{x}_w, \mathbf{x}_l, \mathbf{x}_{\mathrm{prompt}}\} \sim \mathrm{D}_{\mathrm{triplet}}} \Big[ \mathrm{KL}\big( \mathrm{P}_{f_\theta}(.|\mathbf{x}_w, \mathbf{x}_l, \mathbf{x}_{\mathrm{prompt}}) || \mathrm{P}_{r_\phi}(.|\mathbf{x}_w, \mathbf{x}_l, \mathbf{x}_{\mathrm{prompt}}) \big) \Big] \quad (3)$$

### 3.6 INFERENCE AND EVALUATION

During inference, the model starts with an initial seed sequence, iteratively edits it and is expected to improve its fitness. At iteration $t$, given the seed sequence $\mathbf{x}_{t-1}$ and the trained extrapolative protein design model $\mathrm{P}_{\mathrm{triplet}}(.|\mathbf{x})$, one would sample $\mathbf{x}_t \sim \mathrm{P}_{\mathrm{triplet}}(.|\mathbf{x}_{t-1})$ until $t$ reaches $T$ (i.e. 10) predefined iterations. (Fan et al., 2018) introduced "top-k" as fixed number of highest probability vocabulary tokens to keep in inference. (Holtzman et al., 2019) introduced nucleus sampling to avoid degenerate sampling by filtering the unreliable tail of the probability distribution. Particularly, for a given top-p probability, the smallest set of most probable tokens with probabilities that add up to top-p or higher are kept for generation. Temperature ($\tau$) hyper-parameter is used to control the randomness in generation and as a way to balance between greedy search (token with max probability when $\tau \to 0$) and uniform sampling (when $\tau \to \infty$). Inspired from Padmakumar et al. (2023), for each initial seed sequence, we sample N (i.e. 10 for AAV and 2 for GFP) sequences using a combination of top-k and top-p sampling with $k = 10$, $p = 0.95$ and a temperature of 0.7 (1.0) without (with) scorer in inference. At the end of each iteration, we randomly select M (i.e. 10,000 for AAV and 2,000 for GFP) samples from all generated sequences and use them as seeds for next iteration when scorer is not used in the inference. We have only chosen the best sequence out of N generated ones for each seed based on scorer model when used in inference. At the last $T$th iteration, we evaluate the final M samples. For *in-silico* evaluation of GFP and AAV datasets, we used the evaluators trained by Kirjner et al. (2024).

## 4 EXPERIMENTS

### 4.1 DATASETS

In order to assess the extrapolation ability of models on both sequence and fitness landscape, we have utilized the Adeno-associated virus (AAV) and Aequorea victoria GFP (avGFP) datasets processed by Kirjner et al. (2024). They proposed to use *mutational gap*, defined as the minimum number of mutations required from the training set to achieve the optimal fitness, in order to measure the extrapolation ability of protein design models. We used the *medium* and *hard* difficulty split of datasets where mutational gap are 6 and 7 mutations respectively. The characteristics of the datasets are explained in Table 1.

Table 1: Characteristics of datasets for benchmarking

| | GFP | | AAV | |
|---|---|---|---|---|
| | Medium | Hard | Medium | Hard |
| Training region | [1.31,3.02] | [1.30,1.56] | [5.64,7.48] | [4.7,6.42] |
| Extrapolation region | >3.02 | >1.56 | >7.48 | >6.42 |
| Mutational gap (99th) | 6 | 7 | 6 | 7 |
| Number of seeds | 100 | 100 | 100 | 100 |
| Avg. fitness of seeds | 2.28 | 2.28 | 1.49 | 1.49 |
| Avg. fitness of top100 extrapolation sequences | 4.02 | 4.02 | 16.62 | 16.62 |

### 4.2 BENCHMARKED MODELS

When scorer has not been utilized in inference, we compared our proposed method to (i) *Sampling*: unconditional protein design through Prot-T5-XL (Elnaggar et al., 2021) (ii) *Iterative Controlled Extrapolation (ICE)*: extrapolation through learning a local editor by translating proteins with lower fitness to slightly better fitness (Padmakumar et al., 2023) (iii) *Align-plm*: extrapolation via Bradley-Terry (BT) model of ranked proteins with big enough distances (Lee et al., 2023). We could not compare our method against Genhance (Chan et al., 2021) as we couldn't run their code. When scorer has been utilized in inference, we compared with (i) *Iterative sampling*: unconditional protein design through Prot-T5-XL (Elnaggar et al., 2021) with scorer ranking, (ii) *ICE + scorer*: ICE with

scorer ranking, (iii) BiGGS: Gibbs sampling with Graph-based Smoothing in the smoothed fitness landscape (Kirjner et al., 2024), (iv) *LatProtRL*: The recent state-of-the-art model for protein fitness optimization through reinforcement learning in latent space of large language models (Lee et al., 2024).

### 4.3 IMPLEMENTATION DETAILS

We used the CNN models trained by Kirjner et al. (2024) and Dallago et al. (2021) on smoothed fitness landscape of the *training regions* of GFP and AAV datasets respectively and utilized them as scorer functions $f_s$. Following Padmakumar et al. (2023), we created the pairs dataset $D_{\text{pairs}} = \{(\mathbf{x}_1^i, \mathbf{x}_2^i)\}_{i=1}^M$ with $M = 900K(100K)$ training (validation) samples where they follow $|f_s(\mathbf{x}_1) - f_s(\mathbf{x}_2)| < 0.5$ and $\{\mathbf{x}_1, \mathbf{x}_2\}$ can either be from original dataset or masking-infiling (e.g. 5% masking). We trained the local editor model on $D_{\text{pairs}}$ for 10 epochs with the AdamW optimizer (Loshchilov & Hutter, 2017), a learning rate of 1e-4 and batch size of 384.

Next, we created the preference dataset for both proteins following the principles of (i) *Don't go backward* and (ii) *Don't get stuck at the same fitness*. For GFP, we binned sequences based on their smoothed fitness into buckets of [0, 0.25, 0.75, 1, 1.25, 1.5, 1.75, 2, 2.25] and [0, 0.2, 0.4, 0.6, 0.8, 1, 1.2, 1.4, 1.6] for medium and hard difficulty datasets. For AAV, we binned sequences based on their smoothed fitness into buckets of [0, 0.5, 1, 1.5, 2, 2.5, 3, 3.5, 4, 4.5, 5, 5.5, 6, 6.5, 7] and [0.5, 1, 1.5, 2, 2.5, 3, 3.5, 4, 4.5, 5, 5.5, 6, 6.5] for medium and hard difficulty datasets. In total, we created 500K (50K) training (validation) samples in which half of them based on *Don't go backward* and the other half based on *Don't get stuck at the same fitness*. For *Don't go backward*, we sampled triplets from every three consecutive buckets where $\mathbf{x}_{\text{prompts}}$, $\mathbf{x}_l$ and $\mathbf{x}_w$ are from buckets with middle, lowest and highest fitness respectively. For *Don't get stuck at the same fitness*, we sampled triplets from every two consecutive buckets where $\mathbf{x}_{\text{prompts}}$ and $\mathbf{x}_l$ are from the bucket with lower fitness and $\mathbf{x}_w$ is from the bucket with higher fitness. As shown by Padmakumar et al. (2023) in pair creation, utilizing masking-infiling sequences might further improve the triplets as well. Instead of investigating masking-infiling from unconditional generative model for triplets, we have created two preference datasets utilizing paired conditional extrapolative seq2seq model in combination with scorers. These datasets are named *scorer distillation* and *combined distillation* which are explained in details in section 6.1.

Then, we assessed the hardness of triples defined in equation 1. We chose the top 100K (10K) hardest triplets as training (validation) samples for offline preference learning. We further fine-tuned the local editor model based on triplet-based preference learning through EXO loss function defined in equation 3 for 1 epoch with batch size of 32, learning rate of 5e-7, $\beta = 0.1$ and the AdamW optimizer (Loshchilov & Hutter, 2017).

### 4.4 EVALUATION METRICS

We use four evaluation metrics: (i) **Extrapolation percentage**: Our primary goal is to generate sequences in the extrapolation region. We use evaluator models to assess the percentage of generated sequences that have fitness in the extrapolation region. (ii) **Fitness$_{100}$**: Our secondary metric is the average fitness of the top 100 generated sequences that measures how far away generated sequences are from the training region in the fitness landscape. (iii) **Distance$_{100}$**: the third metric measures the edit distance of the top 100 generated sequences from the top 100 ground truth sequences in the actual assay that have not been seen by any model (extrapolation sequences). We measure the average of the closest distance between these two sets of sequences. (iv) **Diversity$_{100}$**: Our final metric measures the diversity of the top 100 candidates of each model. Diversity is measured as median of the distances between every pair of top 100 candidates. We should emphasize that higher diversity does not correlate with better performance since a random algorithm can achieve maximum diversity. However, it may provide insights on the exploitation-exploration trade-off.

## 5 RESULTS

Figure 2 shows that for 3 out of 4 datasets, the proposed method (EXO) significantly outperforms baseline methods while being slightly better on the hard difficulty split of the GFP dataset. Table 2 shows that (i) extrapolation percentage has substantially increased by 1.54 (20.76% to 52.75%),

17.53 (4.59% to 85.07%), 4.07 (18.52% to 92.92%) times on hard difficulty AAV, medium difficulty AAV and medium difficulty GFP datasets respectively in comparison to the best baseline method. It has decreased from 54.45% to 24.27% on the hard difficulty GFP dataset. (ii) The average fitness of the top 100 candidates has increased drastically by 21.53% (9.01 to 10.95), 39.76% (9.43 to 13.18), 10.19% (2.55 to 2.81), 69.03% (2.39 to 4.04) on the hard/medium difficulty splits of the AAV and GFP datasets respectively, in comparison to the best baseline method. (iii) The top 100 generated candidates from EXO are closer to the top 100 unseen ground truth extrapolation sequences in all datasets except being second best on the hard difficulty split of AAV dataset. (iv) The diversity of the sequences generated on the hard splits of AAV and GFP datasets is on-par with prior methods (balance exploitation vs exploration), while having lower diversity on the medium splits of AAV and GFP datasets (more exploitation vs exploration). First, we performed hyper-parameter tuning for $\beta$ since it is used in the preference learning stage. Based on the results shown in Figure 15 and Table 16, $\beta = 0.1$ is outperforming other values on medium splits while performing decent and robust on hard splits. We also performed hyper-parameter tuning for sampling temperature ($\tau$) which is important at inference stage. Similarly, based on Figure 16 and Table 17, $\tau = 0.7$ is outperforming other values on medium splits while performing decent and robust on hard splits. In addition, Figure 13 and Table 14 shows that our results are reproducible by running with five different random seeds for both the hard and the medium splits of AAV and GFP datasets.

Table 2: Comparison of in-silico fitness evaluation for baselines and proposed method on medium/hard difficulty splits of GFP and AAV datasets. We report average (standard deviation) of 5 different runs.

| Method | AAV (hard) | | | | AAV (medium) | | | |
|---|---|---|---|---|---|---|---|---|
| | Extrapolation ↑ | Fitness$_{100}$ ↑ | Distance$_{100}$ ↓ | Diversity$_{100}$ | Extrapolation ↑ | Fitness$_{100}$ ↑ | Distance$_{100}$ ↓ | Diversity$_{100}$ |
| Top 100 ground truth | - | 16.62 | - | 5.01 | - | 16.62 | - | 5.01 |
| Sampling | 1.64%(0.31) | 7.42(0.30) | **4.49(0.35)** | 7.18(0.61) | 1.64%(0.31) | 7.42(0.30) | 4.49(0.35) | 7.18(0.61) |
| ICE | 5.58%(0.04) | 8.18(0.01) | 9.08(0.14) | 13.56(0.18) | 4.59%(0.15) | 9.43(0.04) | 7.72(0.08) | 11.49(0.24) |
| Align-pm | 20.76%(0.00) | 9.01(0.00) | 7.60(0.00) | 8.16(0.00) | 3.49%(0.00) | 8.66(0.00) | 7.29(0.00) | 6.22(0.00) |
| EXO | **52.75%(1.74)** | **10.95(0.17)** | 5.30(0.39) | 8.46(0.27) | **85.07%(1.72)** | **13.18(0.33)** | **1.64(0.41)** | 1.08(0.46) |
| Method | GFP (hard) | | | | GFP (medium) | | | |
| | Extrapolation ↑ | Fitness$_{100}$ ↑ | Distance$_{100}$ ↓ | Diversity$_{100}$ | Extrapolation ↑ | Fitness$_{100}$ ↑ | Distance$_{100}$ ↓ | Diversity$_{100}$ |
| Top 100 ground truth | - | 4.02 | - | 4.64 | - | 4.02 | - | 4.64 |
| Sampling | 18.52%(0.80) | 1.94(0.09) | 10.21(0.50) | 20.10(1.01) | 18.52%(0.80) | 1.94(0.09) | 10.21(0.50) | 20.10(1.01) |
| ICE | 27.16%(0.80) | 2.07(0.01) | 10.93(0.18) | 15.76(0.64) | 0.16%(0.19) | 2.39(0.03) | 8.47(0.21) | 14.41(0.87) |
| Align-plm | **54.45%(0.00)** | 2.55(0.00) | 9.64(0.00) | 4.17(0.00) | 0.00%(0.00) | 2.12(0.00) | 6.13(0.00) | 5.41(0.00) |
| EXO | 24.27%(1.70) | **2.81(0.07)** | **9.08(4.04)** | 14.96(7.44) | **92.92%(0.30)** | **4.04(0.01)** | **2.13(0.14)** | 2.57(0.14) |

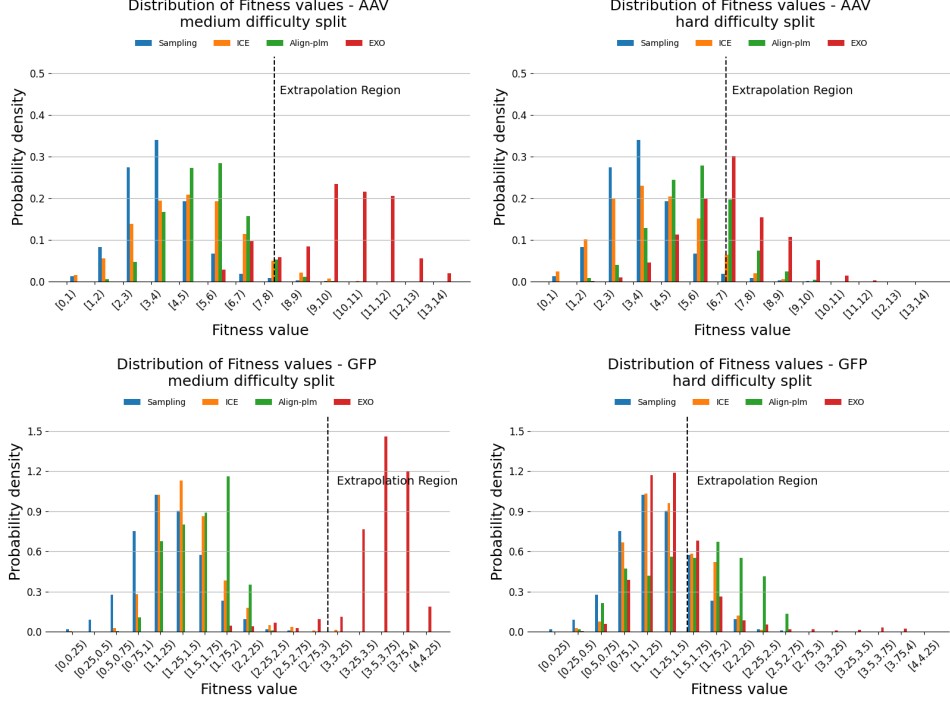

Figure 2: Comparison of in-silico fitness evaluation on GFP and AAV datasets.

# 6 Ablation studies

## 6.1 Effect of preference data

To evaluate the effect of preference data on the proposed method, we have created five types of preference datasets (1) *All*: **random** triplet creation through *Don't go backward* and *Don't get stuck at the same fitness* principles that includes a **mix** of easy and hard examples. (2) *Mistakes*: focusing on the **hardest triplets** created from similar principles (this is the default preference of data creation in the proposed method). (3) *Mistakes in extrapolation*: focusing only on the **hardest triplets close to the extrapolation region** but in the training region. The main question behind it: Are the hard examples close to the extrapolation region sufficient for good extrapolative generative model in extrapolation region? (4) *Scorer distillation*: In the previous three preference datasets, we perform *data distillation* by ranking triplewise sequences based on hardness for local editor. Another alternative approach is to utilize the local editor to propose sequences and rank them based on the scorer in order to create a preference dataset. Particularly, for a given $\mathbf{x}_{\text{prompt}}$, one can sample $K$ sequences $\mathbf{y}_{1:K} = \{\mathbf{y}_1, \mathbf{y}_2, .., \mathbf{y}_K\}$ from $P_{\text{pair}}(.|\mathbf{x}_{\text{prompt}})$. Then, choose the triplewise rankings for all pairs based on $s_{i,j} = f(\mathbf{y}_j) - f(\mathbf{y}_i)$ where $f(\mathbf{y}_j) < f(\mathbf{x}_{\text{prompt}}) < f(\mathbf{y}_i)$. By definition, $s_{ij} > 0$ can be considered as hard example since the pairwise model has wrongly generated $y_j$ with lower predicted fitness than prompt. We would like to encourage the pairwise model to generate more sequences similar to $y_i$ since its predicted fitness is higher than seed. This approach can be considered as *scorer distillation* since we are attempting to distill the scorers knowledge in training and extrapolation region into the generative model. (5) *Combined distillation*: Finally, we combine the hardest triplets from *Mistakes* with *scorer distillation* to utilize the best of both worlds.

Table 3 shows that (1) training on hard examples is drastically beneficial in enhancing the performance of EXO. We observe that in medium difficulty splits of AAV/GFP datasets *Mistakes* significantly outperforms *All* version and perform worse (34.55% to 24.27%) and (54.78% to 52.75%) on hard split of GFP and AAV datasets. In addition, average fitness of top 100 candidates have increased drastically for all datasets except hard split of GFP where it slightly performs worse (2.88 vs 2.81) on hard split of GFP dataset. (2) learning from hard examples close to extrapolation region is not sufficient for EXO. Similarly, we observe that in 3 out of 4 datasets *Mistakes* outperforms *Mistakes in extrapolation* version and perform worse (56.65% to 24.27%) on hard split of GFP, (3) training on *data distillation* outperform *scorer distillation* one for EXO. We observe that in 3 out 4 datasets *Mistakes* outperforms *scorer distillation* version and perform worse (76.89% to 52.75%) on hard split of AAV, and (4) training on *Combined distillation* perform well on hard split of AAV and GFP datasets in comparison to the default one but perform worse on medium split. It shows that in hard difficulty split where there is much less data to learn from it is beneficial to augment data through scorer distillation. However, it is computationally expensive to create triplets through scorer distillation since we need to run local editor in inference for many samples. Table 7 indicates that scorer distillation is more than 200 times more expensive to run on GPU machines in comparison to the default. In general, we can conclude that based on Table 3 and Figure 7, *Mistakes* perform better in comparison to other preference datasets with almost no computational overhead.

Table 3: Comparison of in-silico fitness evaluation for various preference data on medium/hard difficulty splits of GFP/AAV datasets. We report average (standard deviation) of 5 different runs.

| Triplet creation | AAV (hard) | | | | AAV (medium) | | | |
|---|---|---|---|---|---|---|---|---|
| | Extrapolation ↑ | Fitness$_{100}$ ↑ | Distance$_{100}$ ↓ | Diversity$_{100}$ | Extrapolation ↑ | Fitness$_{100}$ ↑ | Distance$_{100}$ ↓ | Diversity$_{100}$ |
| All | 54.78%(3.86) | 7.86(0.00) | 9.99(0.00) | 0.00(0.00) | 31.19%(0.89) | 11.06(0.18) | 6.37(0.16) | 6.08(0.72) |
| Mistakes in extrapolation | 6.72%(0.19) | 8.28(0.11) | 8.66(0.20) | 12.83(0.30) | 8.88%(0.51) | 10.80(0.04) | 5.61(0.27) | 8.75(0.29) |
| Scorer distillation | 76.89%(3.48) | 10.15(0.18) | 7.00(0.00) | 0.00(0.00) | 15.52%(1.47) | 9.83(0.00) | 5.99(0.00) | 0.0(0.00) |
| Combined distillation | **78.19%(2.66)** | **13.97(0.11)** | **1.74(0.25)** | 1.70(0.45) | 70.85%(0.72) | **13.24(0.11)** | 2.85(0.19) | 2.91(0.41) |
| **Mistakes (default)** | 52.75%(1.74) | 10.95(0.17) | 5.30(0.39) | 8.46(0.27) | **85.07%(1.72)** | 13.18(0.33) | **1.64(0.41)** | 1.08(0.46) |

| Triplet creation | GFP (hard) | | | | GFP (medium) | | | |
|---|---|---|---|---|---|---|---|---|
| | Extrapolation ↑ | Fitness$_{100}$ ↑ | Distance$_{100}$ ↓ | Diversity$_{100}$ | Extrapolation ↑ | Fitness$_{100}$ ↑ | Distance$_{100}$ ↓ | Diversity$_{100}$ |
| All | 34.55%(2.22) | 2.88(0.26) | 4.50(1.80) | 6.05(2.99) | 17.65%(2.80) | 3.89(0.01) | 2.03(0.15) | 2.47(0.10) |
| Mistakes in extrapolation | 56.65%(1.18) | 3.49(0.02) | **2.13(0.06)** | 3.16(0.06) | 0.00%(0.00) | 1.99(0.02) | 13.44(2.58) | 17.79(4.28) |
| Scorer distillation | 0.00%(0.00) | 1.35(0.01) | 9.78(0.05) | 0.33(0.06) | 9.85%(5.01) | 3.35(0.28) | 3.70(0.88) | 3.02(0.89) |
| Combined distillation | **65.86%(1.07)** | **3.65(0.02)** | 2.42(0.10) | 4.07(0.08) | 28.84%(2.77) | 3.82(0.02) | 2.24(0.19) | 3.14(0.19) |
| **Mistakes (default)** | 24.27%(1.70) | 2.81(0.07) | 9.08(4.04) | 14.96(7.44) | **92.92%(0.30)** | **4.04(0.01)** | 2.13(0.14) | 2.57(0.14) |

We have further assessed the impact of 1) creating triplets from consecutive vs non-consecutive bins; 2) sequence similarity cutoff between prompt vs desired/undesired responses. Table 4 highlights that naively creating preference data from non-consecutive pairs would confuse the model's learning

and worsen the performance on 3 out of 4 splits. We hypothesize that, in order to properly learn from non-consecutive bins, one needs to develop a more sophisticated prompt to incorporate the distance between bins as well. In addition, Table 4 suggests that max mutation 15 between seed and desired/undesired sequences would slightly improve the performance on hard splits of AAV and GFP datasets while worsen it for medium splits. However, calculating sequence similarity for large dataset is computationally expensive therefore we would not suggest it as our default.

Table 4: Comparison of in-silico fitness evaluation for ablation studies on triplet creation.

| Triplet creation | bins | Max mutations | AAV (hard) | | | | AAV (medium) | | | |
|---|---|---|---|---|---|---|---|---|---|---|
| | | | Extrapolation ↑ | Fitness$_{100}$ ↑ | Distance$_{100}$ ↓ | Diversity$_{100}$ | Extrapolation ↑ | Fitness$_{100}$ ↑ | Distance$_{100}$ ↓ | Diversity$_{100}$ |
| Mistakes | Consecutive | None | 52.75%(1.74) | 10.95(0.17) | 5.30(0.39) | 8.46(0.27) | **85.07%(1.72)** | **13.18(0.33)** | **1.64(0.41)** | 1.08(0.46) |
| Mistakes | Non-Consecutive | None | 32.86%(0.84) | 9.46(0.09) | 7.49(0.36) | 9.02(0.75) | 30.41%(0.35) | 11.06(0.03) | 6.77(0.20) | 7.83(0.30) |
| Mistakes | Consecutive | 5 | 7.24%(0.28) | 8.25(0.04) | 9.00(0.12) | 13.18(0.34) | 10.84%(0.68) | 10.14(0.18) | 7.22(0.10) | 9.06(0.64) |
| Mistakes | Consecutive | 10 | 53.33%(12.83) | 9.58(0.47) | 8.05(0.87) | 2.56(4.32) | 57.47%(0.30) | 10.03(0.01) | 5.99(0.20) | 4.25(0.45) |
| Mistakes | Consecutive | 15 | **55.62%(1.38)** | **11.20(0.29)** | **5.11(0.19)** | 7.37(0.19) | 81.38%(1.86) | 11.72(0.12) | 3.19(0.40) | 2.63(0.27) |

| Triplet creation | bins | Max mutations | GFP (hard) | | | | GFP (medium) | | | |
|---|---|---|---|---|---|---|---|---|---|---|
| | | | Extrapolation ↑ | Fitness$_{100}$ ↑ | Distance$_{100}$ ↓ | Diversity$_{100}$ | Extrapolation ↑ | Fitness$_{100}$ ↑ | Distance$_{100}$ ↓ | Diversity$_{100}$ |
| Mistakes | Consecutive | None | 24.27%(1.70) | **2.81(0.07)** | 9.08(4.04) | 14.96(7.44) | **92.92%(0.30)** | **4.04(0.01)** | 2.13(0.14) | 2.57(0.14) |
| Mistakes | Non-Consecutive | None | 31.61%(4.16) | 2.50(0.09) | 7.08(1.04) | 7.77(1.82) | 82.82%(0.84) | 4.06(0.05) | 0.80(0.40) | 0.69(0.26) |
| Mistakes | Consecutive | 5 | 27.57%(1.42) | 2.05(0.01) | 10.83(0.16) | 14.72(0.65) | 0.05%(0.10) | 2.35(0.05) | 8.68(0.15) | 14.43(0.26) |
| Mistakes | Consecutive | 10 | 11.16%(0.58) | 2.78(0.04) | **3.52(0.30)** | 4.59(0.71) | 3.98%(0.84) | 3.48(0.11) | 3.43(0.32) | 4.50(0.82) |
| Mistakes | Consecutive | 15 | **42.01%(6.84)** | 2.34(0.07) | 6.32(0.21) | 7.51(0.59) | 83.10%(0.38) | 3.95(0.00) | **0.30(0.05)** | **0.00(0.00)** |

## 6.2 EFFECT OF SCORER

Prior to our work, scorer-based generative models such as LatprotRL (through reinforcement learning (Lee et al., 2024)) or BiGGS (through Gibbs sampling (Kirjner et al., 2024)) were the state-of-the-art and outperformed non-scorer generative models by large margins. Tables 5 shows that our method can fill the gap, compete favorably with the scorer-based generative model and outperform them in various tasks. Particularly, *EXO* outperform scorer-based generative model on medium difficulty split of AAV (85.07% vs 38.63%) and GFP (92.92% vs 55.50%) datasets. In addition combination of *EXO* + scorer ranking can furthermore outperform LatproRL/BiGGS in 3 out of 4 datasets. In addition we can observe that, on the hard split of GFP dataset even though extrapolation percentage of *EXO* + *scorer* is 50.91% vs BiGGs 99.53%. However, its average top 100 generated sequences are almost on-par with them (3.79 vs 3.83). In addition its top 100 generated sequences are closer to the unseen extrapolation sequences (1.73 vs 3.48 mutations). Figure 14 and Table 15 indicates that our results are reproducible by running with five different random seed generator for both hard and medium splits of AAV and GFP datasets.

Table 5: Comparison of in-silico fitness evaluation for *scorer-based* methods.

| Method | AAV (hard) | | | | AAV (medium) | | | |
|---|---|---|---|---|---|---|---|---|
| | Extrapolation ↑ | Fitness$_{100}$ ↑ | Distance$_{100}$ ↓ | Diversity$_{100}$ | Extrapolation ↑ | Fitness$_{100}$ ↑ | Distance$_{100}$ ↓ | Diversity$_{100}$ |
| Iterative sampling | 1.37%(0.12) | 7.67(0.25) | 6.54(0.44) | 11.28(0.70) | 0.40%(0.10) | 7.67(0.14) | 6.59(0.65) | 11.43(0.97) |
| ICE + scorer | 37.01%(0.36) | 10.26(0.12) | 6.50(0.30) | 9.90(0.47) | 33.17%(0.59) | 10.80(0.10) | 6.52(0.49) | 9.89(0.55) |
| BiGGS | 16.80%(5.37) | 10.85(0.51) | 5.70(1.06) | 6.38(1.44) | 4.88%(0.84) | 10.21(0.88) | 8.05(0.84) | 8.34(0.93) |
| LatprotRL | 64.82%(1.02) | 13.29(0.06) | 2.45(0.16) | 4.67(0.23) | 38.63%(0.86) | 12.53(0.08) | 2.83(0.15) | 5.21(0.07) |
| **EXO** | 52.75%(1.74) | 10.95(0.17) | 5.30(0.39) | 8.46(0.27) | 85.07%(1.72) | 13.18(0.33) | **1.64(0.41)** | 1.08(0.46) |
| **EXO + scorer** | 84.04%(1.15) | **14.25(0.11)** | 1.52(0.21) | 2.53(0.31) | **94.23%(0.60)** | **13.90(0.05)** | 2.00(0.06) | 3.05(0.29) |
| **EXO (Combined distillation)** | 78.19%(2.66) | 13.97(0.11) | 1.74(0.25) | 1.70(0.45) | 70.85%(0.72) | 13.24(0.11) | 2.85(0.19) | 2.91(0.41) |
| **EXO (Combined distillation) + scorer** | **98.96%(0.21)** | 14.21(0.02) | **1.51(0.23)** | 1.08(0.19) | 84.71%(0.39) | 13.19(0.03) | 3.14(0.16) | 3.34(0.28) |

| Method | GFP (hard) | | | | GFP (medium) | | | |
|---|---|---|---|---|---|---|---|---|
| | Extrapolation ↑ | Fitness$_{100}$ ↑ | Distance$_{100}$ ↓ | Diversity$_{100}$ | Extrapolation ↑ | Fitness$_{100}$ ↑ | Distance$_{100}$ ↓ | Diversity$_{100}$ |
| Iterative sampling | 0.01%(0.02) | 1.18(0.03) | 184.71(10.78) | 207.45(3.02) | 0.00%(0.00) | 1.22(0.04) | 174.52(17.62) | 201.22(6.04) |
| ICE + scorer | 14.87%(0.75) | 1.96(0.04) | 9.52(0.25) | 18.19(0.46) | 0.02%(0.02) | 2.53(0.04) | 8.22(0.14) | 15.68(0.28) |
| BiGGS | **99.53%(0.21)** | 3.83(0.02) | 3.48(0.36) | 6.01(0.51) | 55.50%(6.75) | 3.89(0.03) | 4.13(0.38) | 5.74(0.71) |
| LatprotRL | 88.28%(1.05) | **3.88(0.01)** | **1.48(0.04)** | 2.86(0.07) | 38.22%(1.99) | 3.92(0.01) | **1.56(0.05)** | 3.04(0.05) |
| **EXO** | 24.27%(1.70) | 2.81(0.07) | 9.08(4.04) | 14.96(7.44) | **92.92%(0.30)** | **4.04(0.01)** | 2.13(0.14) | 2.57(0.14) |
| **EXO + scorer** | 50.91%(3.16) | 3.79(0.01) | 1.73(0.12) | 3.08(0.22) | 58.09%(6.35) | 3.96(0.03) | 2.75(0.16) | 4.04(0.10) |
| **EXO (Combined distillation)** | 65.86%(1.07) | 3.65(0.02) | 2.42(0.10) | 4.07(0.08) | 28.84%(2.77) | 3.82(0.02) | 2.24(0.19) | 3.14(0.19) |
| **EXO (Combined distillation) + scorer** | 71.15%(2.34) | 3.75(0.03) | 2.10(0.07) | 3.46(0.09) | 32.41%(1.34) | 3.86(0.02) | 2.16(0.11) | 3.19(0.16) |

## 6.3 EFFECT OF PREFERENCE LEARNING ALGORITHM

We benchmark against the the state-of-the-art offline preference learning algorithms such as direct preference optimization (DPO) (Rafailov et al., 2023), its generalized version of identity-mapping preference optimization (IPO) (Azar et al., 2023), addition of NLL loss function on favorable direction to DPO (IRPO) (Pang et al., 2024), noise contrastive alignment (NCA) (Chen et al., 2024), Nash equilibrium based self-play preference optimization (SPPO) (Wu et al., 2024), alignment through optimal transport (AOT) (Melnyk et al., 2024) and our default model EXO through reverse KL distribution matching (Ji et al., 2024). Tables 6 shows that EXO outperforms other preference learning algorithms on 3 out of 4 tasks. Particularly, *EXO* outperform the best alternative preference

learning algorithm by (52.75% vs 50.78% attained by AOT), (85.07% vs 74.39% attained by DPO) and (92.92% vs 47.26% attained by IRPO) on hard split of AAV, medium split of AAV and medium split of GFP, respectively. IRPO through addition of NLL loss function to DPO loss has shown to outperforms others on hard split of GFP by 55.15% vs 42.80% attained by IPO. In addition, training curves for preference learning based on these preference learning methods have been shown on Figures 17, 18, 20, 19. In general, we can observe that *EXO* is robustly performing well.

Table 6: Comparison of in-silico fitness evaluation for various preference learning algorithms.

| Method | AAV (hard) | | | | AAV (medium) | | | |
|---|---|---|---|---|---|---|---|---|
| | Extrapolation $\uparrow$ | Fitness$_{100}$ $\uparrow$ | Distance$_{100}$ $\downarrow$ | Diversity$_{100}$ | Extrapolation $\uparrow$ | Fitness$_{100}$ $\uparrow$ | Distance$_{100}$ $\downarrow$ | Diversity$_{100}$ |
| DPO | 46.53%(0.53) | **11.17(0.12)** | 5.68(0.17) | 9.15(0.34) | 74.39%(2.41) | 12.96(0.09) | 2.77(0.45) | 4.15(0.33) |
| IPO | 33.74%(0.81) | 9.97(0.15) | 7.20(0.19) | 11.15(0.13) | 30.83%(0.34) | 11.47(0.08) | 5.41(0.10) | 7.30(0.14) |
| IRPO | 37.67%(0.76) | 10.62(0.29) | 5.36(0.31) | 8.35(0.29) | 35.13%(0.65) | 11.37(0.10) | 5.39(0.22) | 7.86(0.45) |
| NCA | 39.10%(1.89) | 10.07(0.23) | 6.90(0.56) | 10.38(0.43) | 39.75%(1.10) | 11.95(0.29) | 4.19(0.55) | 5.92(0.81) |
| SPPO | 21.31%(0.56) | 9.18(0.07) | 8.22(0.08) | 11.30(0.40) | 17.54%(2.25) | 10.88(0.10) | 6.57(0.09) | 7.97(0.11) |
| AOT | 50.78%(1.55) | 10.97(0.08) | 5.79(0.62) | 9.31(0.58) | 59.35%(1.91) | 12.83(0.09) | 2.68(0.45) | 3.61(0.94) |
| **EXO** | **52.75%(1.74)** | 10.95(0.17) | **5.30(0.39)** | 8.46(0.27) | **85.07%(1.72)** | **13.18(0.33)** | **1.64(0.41)** | 1.08(0.46) |

| Method | GFP (hard) | | | | GFP (medium) | | | |
|---|---|---|---|---|---|---|---|---|
| | Extrapolation $\uparrow$ | Fitness$_{100}$ $\uparrow$ | Distance$_{100}$ $\downarrow$ | Diversity$_{100}$ | Extrapolation $\uparrow$ | Fitness$_{100}$ $\uparrow$ | Distance$_{100}$ $\downarrow$ | Diversity$_{100}$ |
| DPO | 38.96%(5.99) | 3.73(0.06) | 1.92(0.19) | 3.26(0.28) | 13.76%(4.64) | 3.53(0.12) | 3.77(0.60) | 5.05(0.51) |
| IPO | 42.80%(9.05) | 3.47(0.08) | 2.05(0.11) | 3.10(0.24) | 10.55%(0.33) | 3.69(0.03) | 2.45(0.09) | 4.00(0.19) |
| IRPO | **55.15%(1.94)** | **3.68(0.06)** | **1.85(0.44)** | 2.94(0.54) | 47.26%(1.14) | 3.85(0.01) | **2.10(0.03)** | 3.68(0.08) |
| NCA | 30.75%(1.01) | 2.55(0.09) | 5.65(0.97) | 9.08(1.57) | 24.67%(1.75) | 3.92(0.02) | 2.32(0.05) | 2.98(0.14) |
| SPPO | 30.70%(1.60) | 2.32(0.07) | 9.07(0.37) | 12.03(1.05) | 0.58%(0.12) | 2.51(0.07) | 7.29(0.19) | 11.63(0.68) |
| AOT | 35.51%(5.60) | 3.38(0.09) | 9.90(4.50) | 18.15(8.12) | 29.54%(5.40) | 3.65(0.03) | 3.95(0.12) | 4.84(0.22) |
| **EXO** | 24.27%(1.70) | 2.81(0.07) | 9.08(4.04) | 14.96(7.44) | **92.92%(0.30)** | **4.04(0.01)** | 2.13(0.14) | 2.57(0.14) |

# 7 DISCUSSIONS

We utilized the ProstT5 model (Heinzinger et al., 2023) trained in multi-modal fashion (sequence and structure) to embed the unique sequences generated by each method. ProstT5 would enable us to better assess the closeness of the top generated candidates to the top unseen extrapolation sequences in sequence-structure latent space rather than defining closeness based on edit distance in purely sequence space, which does not consider the characteristics of mutations. Two dimensional visualization of embeddings through t-SNE (Van der Maaten & Hinton, 2008) in Figure 3 highlights that top candidates generated by EXO and EXO+scorer have supports by unseen ground truth extrapolation sequences in the latent space of ProstT5 on the hard split of AAV dataset.

As mentioned, we used the in-silico evaluator trained on GFP and AAV datasets by Kirjner et al. (2024) and utilized by the state-of-the-art methods (Lee et al., 2024; 2023). We calculated the mean-squared error and Spearman's rank correlation coefficient between a scorer trained on a specific split of the dataset and the evaluator trained on the entire dataset. Figures 4 and 5 highlights that (1) the mean square error of scorer deteriorates significantly in the extrapolation region. (2) on the AAV dataset the correlation deteriorates as we move further away from the training region, as expected. However, on the GFP dataset, the scorer and evaluator have a very non-smooth rank correlation relationship. An experimental pipeline with wet-lab validation in the loop is the ultimate evaluation pipeline which is, unfortunately, very costly and time-intensive.

# 8 CONCLUSION

We present a novel *data distillation* approach to enhance extrapolative protein design models. Our main contribution is to create model-aware hard preference datasets in order to better approximate the direction of the gradient and learn higher order relationships such as triplewise rankings. In order to evaluate our contribution, we have utilized the carefully curated medium/hard difficulty splits of AAV and GFP datasets where the model needed to extrapolate both on sequence and fitness spaces. Our framework outperforms both extrapolative non-scorer and scorer-based generative models baselines. Through ablation studies, we have investigated (1) the effects of various approaches of creating preference datasets and (2) effects of state-of-the-art preference learning algorithms. Potential future directions include (1) assessing the effect of higher order relationship (quadruples etc.) through Plackett-Luce ranking models (Plackett, 1975; Luce, 2005) on extrapolation, and (2) utilizing reasoning approaches such as tree of thoughts (Yao et al., 2024) to boost performance of the proposed extrapolative protein design model.

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

## A    GENERAL MODEL ALIGNMENT

Ji et al. (2024) proposed the following definition for general model alignment:

**Definition A.1.** Let $\beta_\pi > 0$, $\beta_r > 0$ and $\beta_\pi \beta_r = \beta$. In addition for simplicity, let's assume $\mathbf{x}$ is the prompt sequence, $\mathbf{y}$ is the desired or undesired sequence and $r_\phi(\mathbf{x}, \mathbf{y})$ is the reward model between prompt and generated sequence. Therefore, the generalized alignment objective is defined as:

$$\mathcal{J} = \mathbb{E}_{\mathbf{x} \sim D_{\text{triplet}}} \left( \mathbb{E}_{P_\theta^{\beta_\pi}(\mathbf{y}|\mathbf{x})}[r_\phi(\mathbf{x}, \mathbf{y})] - \beta_r D_{\text{KL}}[P_\theta^{\beta_\pi}(\mathbf{y}|\mathbf{x})||P_{\text{pair}}(\mathbf{y}|\mathbf{x})] \right), \tag{4}$$

Where $P_\theta^{\beta_\pi}(\mathbf{y}|\mathbf{x})$ satisfies:

$$P_\theta^{\beta_\pi}(\mathbf{y}|\mathbf{x}) \propto P_\theta(\mathbf{y}|\mathbf{x})^{\beta_\pi} P_{\text{pair}}(\mathbf{y}|\mathbf{x})^{1-\beta_\pi}, \tag{5}$$

In addition, given unlimited model capacity, the optimal policy is analytically defined as:

$$P_\beta^* = P_{\text{pair}}(\mathbf{y}|\mathbf{x}) \frac{e^{\frac{1}{\beta} r_\phi(\mathbf{x},\mathbf{y})}}{Z_\beta(\mathbf{x})}, \tag{6}$$

Where $Z_\beta(\mathbf{x}) = \sum_{\mathbf{y}' \sim \mathcal{Y}} P_{\text{pair}}(\mathbf{y}'|\mathbf{x}) e^{\frac{1}{\beta} r_\phi(\mathbf{x},\mathbf{y}')}$ is the partition function.

Ji et al. (2024) have proposed a practical way of approximating the general alignment objective through sampling and self-normalization. Following their proposed approach, for given S generated sequences $\mathbf{y}_{1:S} = \{\mathbf{y}_1, \mathbf{y}_2, .., \mathbf{y}_S\}$ sampled from $P_{\text{pair}}(.|\mathbf{x})$, one can define the empirical distribution $f_\theta$ based on S samples as:

$$P_{f_\theta}(s|\mathbf{y}_{1:S}, \mathbf{x}) = \frac{e^{\beta_\pi \log \frac{P_\theta(\mathbf{y}_s|\mathbf{x})}{P_{\text{pair}}(\mathbf{y}_s|\mathbf{x})}}}{\sum_j e^{\beta_\pi \log \frac{P_\theta(\mathbf{y}_j|\mathbf{x})}{P_{\text{pair}}(\mathbf{y}_j|\mathbf{x})}}} \tag{7}$$

Similarly one can define the empirical distribution $r_\phi$ based on reward function as:

$$P_{r_\phi}(s|\mathbf{y}_{1:S}, \mathbf{x}) = \frac{e^{\frac{1}{\beta_r} r_\phi(\mathbf{x},\mathbf{y}_s)}}{\sum_j e^{\frac{1}{\beta_r} r_\phi(\mathbf{x},\mathbf{y}_j)}} \tag{8}$$

Finally, the original general alignment objective can be translated to the reverse KL between $P_{f_\theta}$ and $P_{r_\phi}$:

$$\mathcal{L}_{\text{exo}} = \mathbb{E}_{\mathbf{x} \sim D_{\text{triplet}}, \mathbf{y}_{1:S} \sim \text{pair}(.|\mathbf{x})} \left[ \text{KL}\big(P_{f_\theta}(.|\mathbf{y}_{1:S}, \mathbf{x})||P_{r_\phi}(.|\mathbf{y}_{1:S}, \mathbf{x})\big) \right] \tag{9}$$

## B    MUTATIONAL ANALYSIS

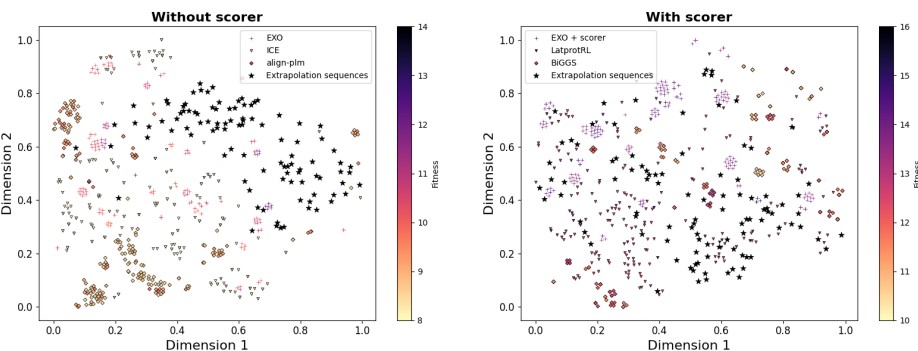

Figure 3: t-SNE visualization (2 dimensions) of top 200 generated sequences for EXO vs baselines based on ProstT5 embedding (Heinzinger et al., 2023) (left) without scorer generative models (right) with scorer generative models on the hard split of AAV dataset.

# C  SCORER VS EVALUATOR

## C.1  ABSOLUTE PREDICTIONS

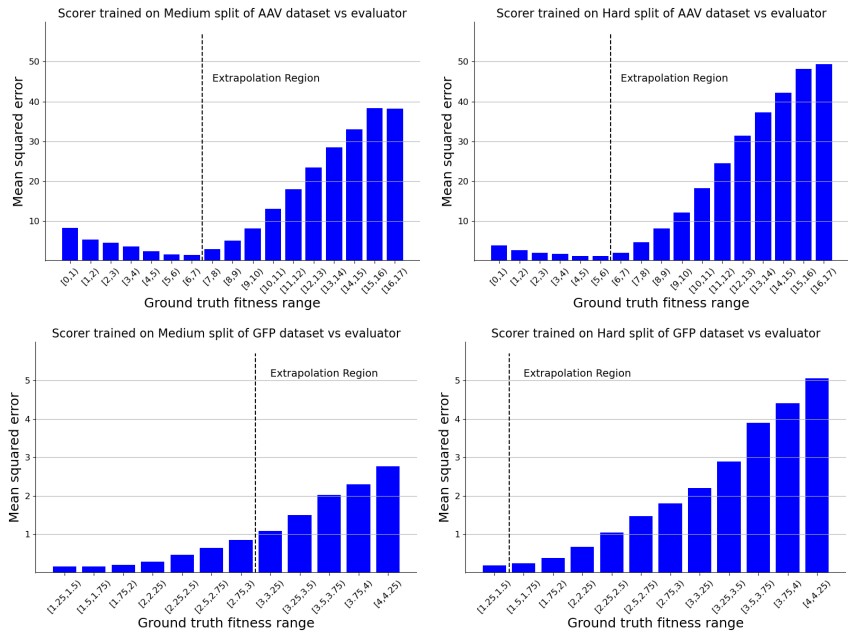

Figure 4: Comparison of in-silico scorer trained on specific subset of the data versus in-silico evaluator trained on all the data from absolute prediction perspective.

## C.2  CORRELATION WITHIN BUCKETS

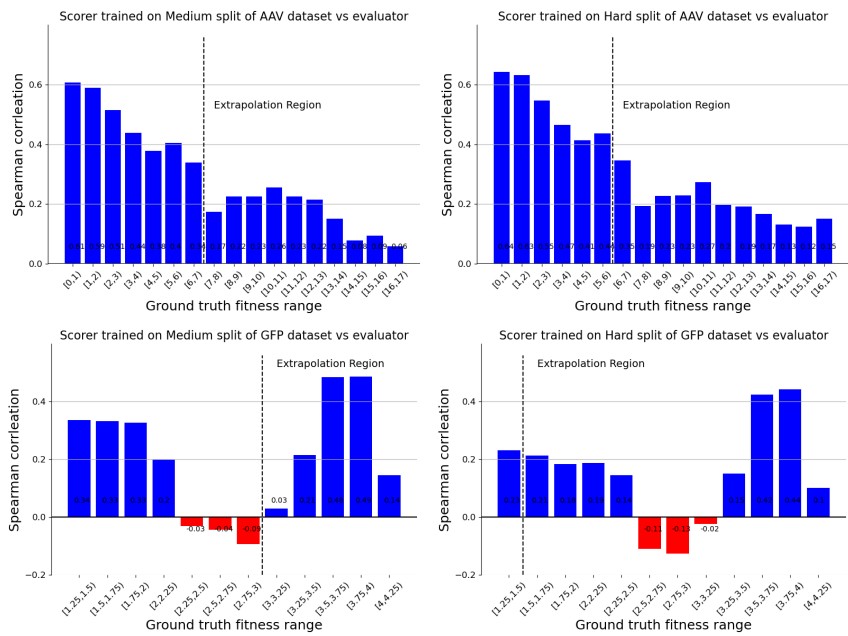

Figure 5: Comparison of in-silico scorer trained on specific subset of the data versus in-silico evaluator trained on all the data from ranking within buckets perspective.

## C.3 RANKING ACROSS BUCKETS

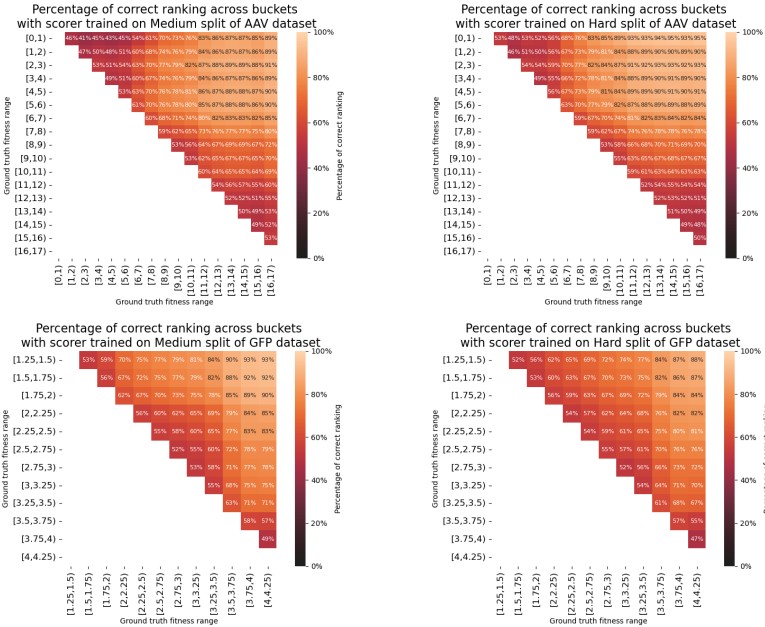

Figure 6: Comparison of in-silico scorer trained on specific subset of the data versus in-silico evaluator trained on all the data from ranking across buckets perspective.

# D ABLATION STUDIES

## D.1 TRIPLETS

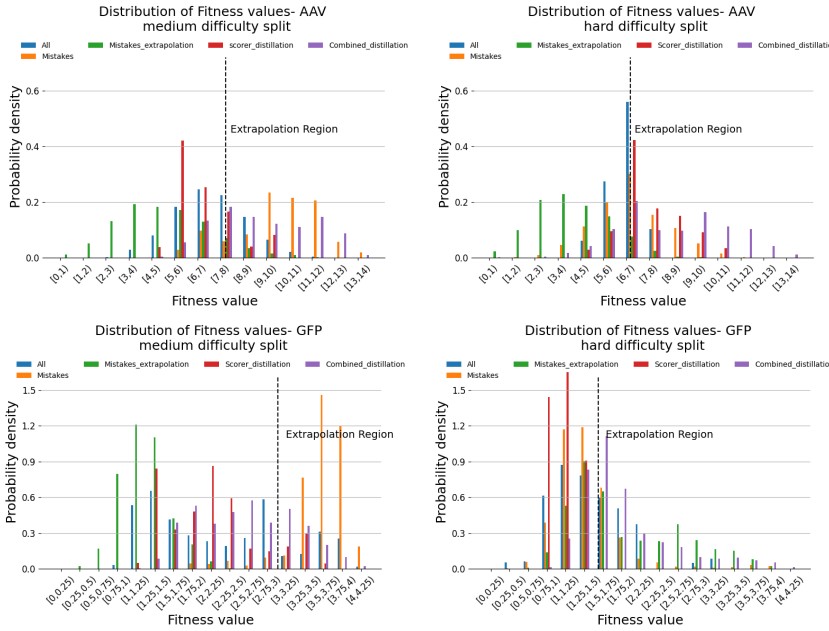

Figure 7: Comparison of in-silico fitness evaluation for various preference data on medium/hard difficulty splits of GFP and AAV datasets.

## D.2 EFFECT OF SCORER

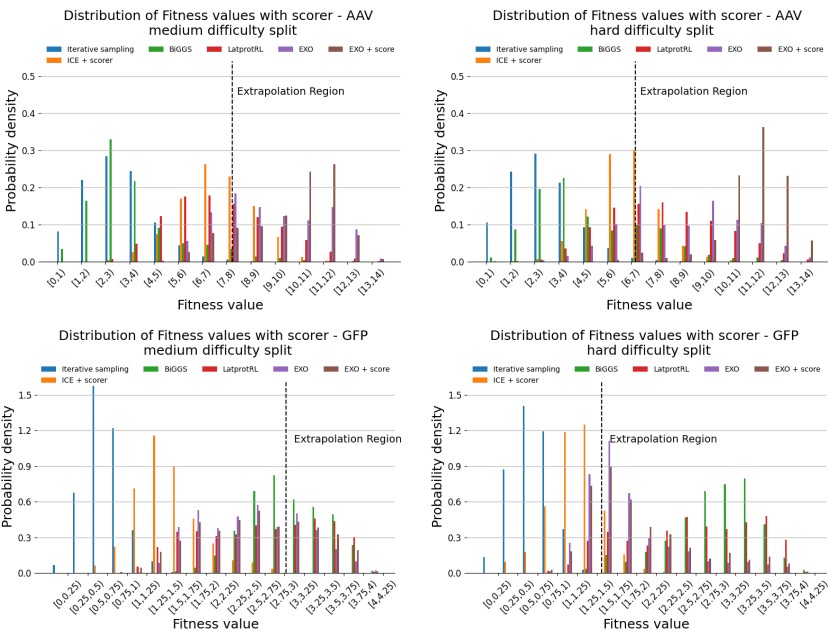

Figure 8: Comparison of in-silico fitness evaluation for *scorer-based* baselines and proposed method on medium/hard difficulty splits of GFP and AAV datasets.

## D.3 EFFECT OF PREFERENCE LEARNING

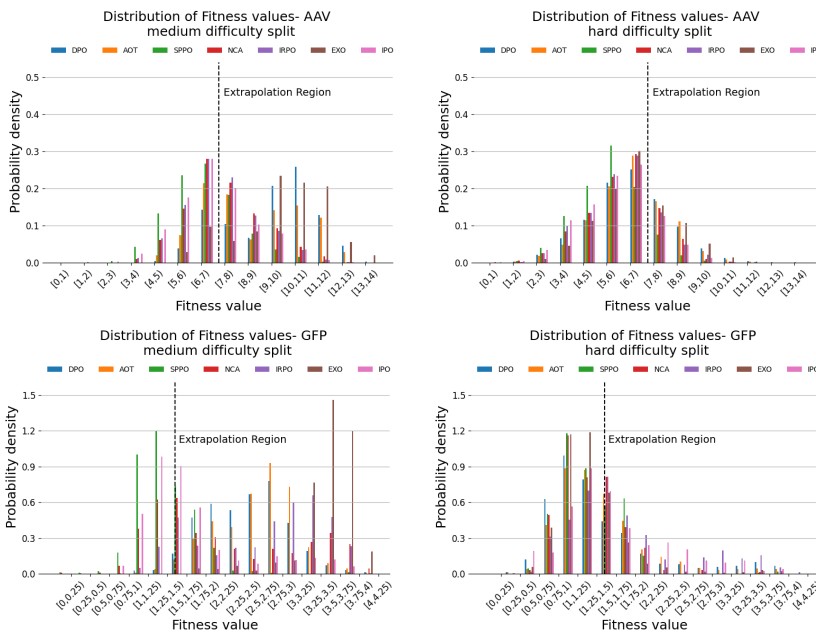

Figure 9: Comparison of in-silico fitness evaluation for various preference learning algorithm on medium/hard difficulty splits of GFP and AAV datasets.

Table 7: Comparison of computational costs of generating triplets with P3 (V100) GPU machine. We report average (standard deviation) of seconds needed for one triplet creation.

| Triplet creation | AAV (seconds) | GFP (seconds) |
|---|---|---|
| **Mistakes** | 0.025(0.001) | 0.031(0.001) |
| **Scorer distillation** | 0.136(0.001) | 8.944(0.007) |

## D.4 CONSECUTIVE VS NON-CONSECUTIVE TRIPLETS

### D.4.1 WITHOUT SCORER

Table 8: Comparison of in-silico fitness evaluation for preference data created based on consecutive or non-consecutive bins on medium/hard difficulty splits of GFP and AAV datasets. We report average (standard deviation) of 5 different runs.

| Triplet creation | AAV (hard) | | | | AAV (medium) | | | |
|---|---|---|---|---|---|---|---|---|
| | Extrapolation $\uparrow$ | $Fitness_{100} \uparrow$ | $Distance_{100} \downarrow$ | $Diversity_{100}$ | Extrapolation $\uparrow$ | $Fitness_{100} \uparrow$ | $Distance_{100} \downarrow$ | $Diversity_{100}$ |
| **Consecutive** | **52.75%(1.74)** | **10.95(0.17)** | **5.30(0.39)** | 8.46(0.27) | **85.07%(1.72)** | **13.18(0.33)** | **1.64(0.41)** | 1.08(0.46) |
| **Non-Consecutive** | 32.86%(0.84) | 9.46(0.09) | 7.49(0.36) | 9.02(0.75) | 30.41%(0.35) | 11.06(0.03) | 6.77(0.20) | 7.83(0.30) |

| Triplet creation | GFP (hard) | | | | GFP (medium) | | | |
|---|---|---|---|---|---|---|---|---|
| | Extrapolation $\uparrow$ | $Fitness_{100} \uparrow$ | $Distance_{100} \downarrow$ | $Diversity_{100}$ | Extrapolation $\uparrow$ | $Fitness_{100} \uparrow$ | $Distance_{100} \downarrow$ | $Diversity_{100}$ |
| **Consecutive** | 24.27%(1.70) | **2.81(0.07)** | 9.08(4.04) | 14.96(7.44) | **92.92%(0.30)** | 4.04(0.01) | 2.13(0.14) | 2.57(0.14) |
| **Non-Consecutive** | **31.61%(4.16)** | 2.50(0.09) | **7.08(1.04)** | 7.77(1.82) | 82.82%(1.56) | 4.06(0.05) | **0.80(0.40)** | 0.69(0.26) |

### D.4.2 WITH SCORER

Table 9: Comparison of in-silico fitness evaluation for preference data created based on consecutive or non-consecutive bins on medium/hard difficulty splits of GFP and AAV datasets with scorer. We report average (standard deviation) of 5 different runs.

| Triplet creation | AAV (hard) | | | | AAV (medium) | | | |
|---|---|---|---|---|---|---|---|---|
| | Extrapolation $\uparrow$ | $Fitness_{100} \uparrow$ | $Distance_{100} \downarrow$ | $Diversity_{100}$ | Extrapolation $\uparrow$ | $Fitness_{100} \uparrow$ | $Distance_{100} \downarrow$ | $Diversity_{100}$ |
| **Consecutive + scorer** | **84.04%(1.15)** | **14.25(0.11)** | **1.52(0.21)** | 2.53(0.31) | **94.23%(0.60)** | **13.90(0.05)** | **2.00(0.06)** | 3.05(0.29) |
| **Non-Consecutive + scorer** | 54.92%(0.53) | 10.65(0.18) | 6.12(0.34) | 8.56(0.63) | 44.9%(0.80) | 11.43(0.12) | 5.51(0.35) | 7.75(0.43) |

| Triplet creation | GFP (hard) | | | | GFP (medium) | | | |
|---|---|---|---|---|---|---|---|---|
| | Extrapolation $\uparrow$ | $Fitness_{100} \uparrow$ | $Distance_{100} \downarrow$ | $Diversity_{100}$ | Extrapolation $\uparrow$ | $Fitness_{100} \uparrow$ | $Distance_{100} \downarrow$ | $Diversity_{100}$ |
| **Consecutive +scorer** | **50.91%(3.16)** | **3.79(0.01)** | **1.73(0.12)** | 3.08(0.22) | 58.09%(6.35) | **3.96(0.03)** | 2.75(0.16) | 4.04(0.10) |
| **Non-Consecutive + scorer** | 35.46%(2.97) | 2.91(0.09) | 3.23(0.47) | 5.29(0.54) | **68.76%(12.84)** | **3.96(0.06)** | **1.81(0.29)** | 2.28(0.26) |

## D.5 SEQUENCE SIMILARITY CONSTRAINT

### D.5.1 DISTRIBUTION OF SEQUENCE SIMILARITY IN CURRENT PREFERENCE DATA

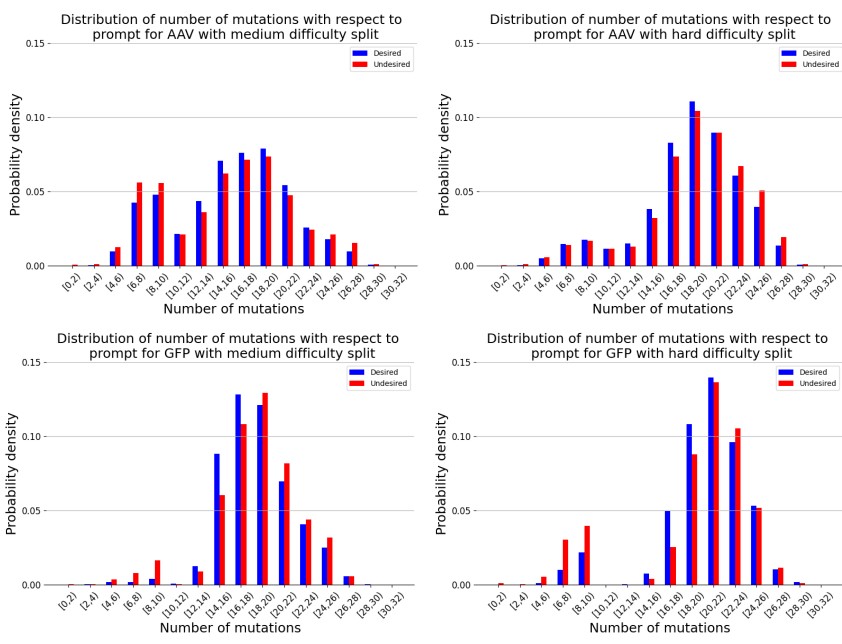

Figure 10: Distribution of sequence similarities on hard triplets of AAV and GFP datasets.

### D.5.2 CORRELATION OF SEQUENCE SIMILARITY IN PREFERENCE DATA WITH ITS HARDNESS

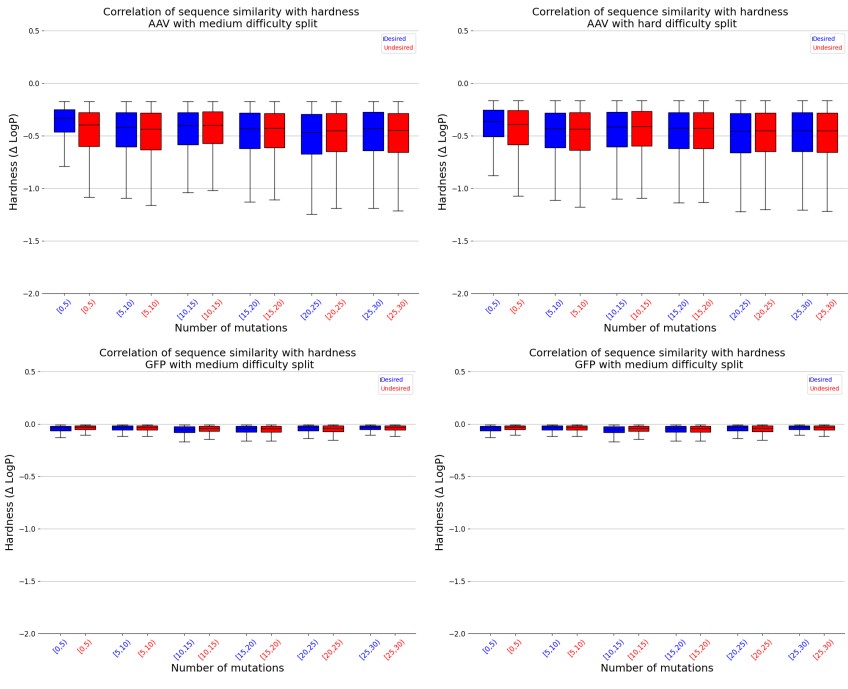

Figure 11: Sequence similarity does not have correlation with its hardness in triplets on both AAV and GFP datasets.

### D.5.3 WITHOUT SCORER

Table 10: Comparison of in-silico fitness evaluation for sequence similarity cutoff preference dataset on medium/hard difficulty splits of GFP and AAV datasets. We report average (standard deviation) of 5 different runs.

| Triplet creation | AAV (hard) | | | | AAV (medium) | | | |
|---|---|---|---|---|---|---|---|---|
| | Extrapolation ↑ | Fitness$_{100}$ ↑ | Distance$_{100}$ ↓ | Diversity$_{100}$ | Extrapolation ↑ | Fitness$_{100}$ ↑ | Distance$_{100}$ ↓ | Diversity$_{100}$ |
| max mutation = 5 | 7.24%(0.28) | 8.25(0.04) | 9.00(0.12) | 13.18(0.34) | 10.84%(0.68) | 10.14(0.18) | 7.22(0.10) | 9.06(0.64) |
| max mutation = 10 | 53.33%(12.83) | 9.58(0.47) | 8.05(0.87) | 2.56(4.32) | 57.47%(0.30) | 10.03(0.01) | 5.99(0.20) | 4.25(0.45) |
| max mutation = 15 | **55.62%(1.38)** | **11.20(0.29)** | **5.11(0.19)** | 7.37(0.19) | 81.38%(1.86) | 11.72(0.12) | 3.19(0.40) | 2.63(0.27) |
| No max mutation constraint | 52.75%(1.74) | 10.95(0.17) | 5.30(0.39) | 8.46(0.27) | **85.07%(1.72)** | **13.18(0.33)** | **1.64(0.41)** | 1.08(0.46) |

| Triplet creation | GFP (hard) | | | | GFP (medium) | | | |
|---|---|---|---|---|---|---|---|---|
| | Extrapolation ↑ | Fitness$_{100}$ ↑ | Distance$_{100}$ ↓ | Diversity$_{100}$ | Extrapolation ↑ | Fitness$_{100}$ ↑ | Distance$_{100}$ ↓ | Diversity$_{100}$ |
| max mutation = 5 | 27.57%(1.42) | 2.05(0.01) | 10.83(0.16) | 14.72(0.65) | 2.35(0.05) | 8.68(0.15) | 14.43(0.26) | |
| max mutation = 10 | 11.16%(0.58) | 2.78(0.04) | **3.52(0.30)** | 4.59(0.71) | 3.98%(0.84) | 3.48(0.11) | 3.43(0.32) | 4.50(0.82) |
| max mutation = 15 | **42.01%(6.84)** | 2.34(0.07) | 6.32(0.21) | 7.51(0.59) | 83.10%(0.38) | 3.95(0.00) | **0.30(0.05)** | 0.00(0.00) |
| No max mutation constraint | 24.27%(1.70) | **2.81(0.07)** | 9.08(4.04) | 14.96(7.44) | **92.92%(0.30)** | **4.04(0.01)** | 2.13(0.14) | 2.57(0.14) |

### D.5.4 WITH SCORER

Table 11: Comparison of in-silico fitness evaluation for sequence similarity cutoff preference dataset on medium/hard difficulty splits of GFP and AAV datasets. We report average (standard deviation) of 5 different runs.

| Triplet creation | AAV (hard) | | | | AAV (medium) | | | |
|---|---|---|---|---|---|---|---|---|
| | Extrapolation ↑ | Fitness$_{100}$ ↑ | Distance$_{100}$ ↓ | Diversity$_{100}$ | Extrapolation ↑ | Fitness$_{100}$ ↑ | Distance$_{100}$ ↓ | Diversity$_{100}$ |
| max mutation = 5 + scorer | 37.14%(1.47) | 9.23(0.06) | 7.91(0.46) | 8.29(1.30) | 32.15(0.29) | 10.76(0.09) | 6.39(0.27) | 8.61(0.45) |
| max mutation = 10 + scorer | **97.14%(5.70)** | 9.46(0.24) | 8.00(0.15) | 0.57(0.28) | 95.34%(0.67) | 13.56(0.09) | **1.54(0.15)** | 2.39(0.22) |
| max mutation = 15 + scorer | 92.97%(1.91) | **14.37(0.06)** | **1.45(0.17)** | 2.32(0.12) | 95.01% (0.75) | 13.81(0.06) | 1.89(0.22) | 3.00(0.37) |
| No max mutation constraint + scorer | 84.04%(1.15) | 14.25(0.11) | 1.52(0.21) | 2.53(0.31) | **94.23%(0.60)** | **13.90(0.05)** | 2.00(0.06) | 3.05(0.29) |

| Triplet creation | GFP (hard) | | | | GFP (medium) | | | |
|---|---|---|---|---|---|---|---|---|
| | Extrapolation ↑ | Fitness$_{100}$ ↑ | Distance$_{100}$ ↓ | Diversity$_{100}$ | Extrapolation ↑ | Fitness$_{100}$ ↑ | Distance$_{100}$ ↓ | Diversity$_{100}$ |
| max mutation = 5 + scorer | 20.81%(1.51) | 2.06(0.02) | 10.91(0.19) | 15.40(1.03) | 0.60%(0.14) | 2.61(0.09) | 8.15(0.08) | 13.99(0.58) |
| max mutation = 10 + scorer | 21.67%(2.76) | 3.10(0.02) | 3.02(0.09) | 4.15(0.22) | 6.18%(0.82) | 3.66(0.05) | 3.08(0.21) | 3.90(0.70) |
| max mutation = 15 + scorer | 41.31%(2.10) | 2.71(0.09) | 4.63(0.07) | 6.49(0.27) | **64.35%(7.20)** | 3.93(0.04) | 2.95(0.11) | 3.04(0.12) |
| No max mutation constraint + scorer | **50.91%(3.16)** | **3.79(0.01)** | **1.73(0.12)** | 3.08(0.22) | 58.09%(6.35) | **3.96(0.03)** | **2.75(0.16)** | 4.04(0.10) |

### D.6 EFFECT OF NUMBER OF ITERATIONS

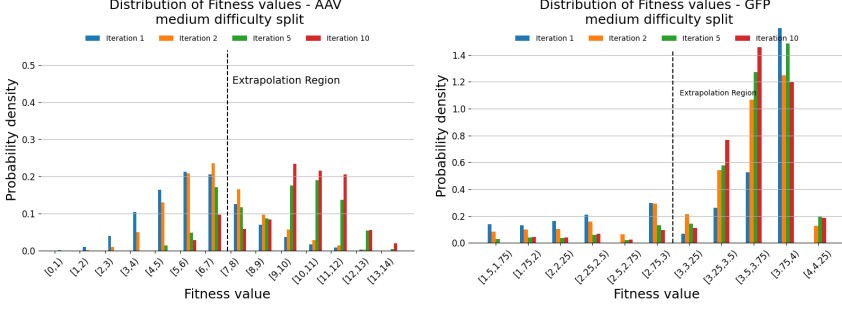

Figure 12: Comparison of in-silico fitness evaluation for various iterations of EXO on medium difficulty split of GFP and AAV datasets.

Table 12: Comparison of in-silico fitness evaluation for various iterations of EXO on medium difficulty split of GFP and AAV datasets.

| Iteration | AAV (medium) | | | | GFP (medium) | | | |
|---|---|---|---|---|---|---|---|---|
| | Extrapolation ↑ | Fitness$_{100}$ ↑ | Distance$_{100}$ ↓ | Diversity$_{100}$ | Extrapolation ↑ | Fitness$_{100}$ ↑ | Distance$_{100}$ ↓ | Diversity$_{100}$ |
| 1 | 18.98% | 11.75 | 3.68 | 6.13 | 76.5% | 3.96 | **0.88** | 0.78 |
| 2 | 26.87% | 11.99 | 3.46 | 5.46 | 80.05% | 4.02 | 1.82 | 1.49 |
| 5 | 69.17% | 13.01 | 1.67 | 1.95 | 92.05% | **4.05** | 2.44 | 2.47 |
| 10 | **83.83%** | **13.50** | **1.00** | 0.99 | **93.10%** | 4.04 | 1.93 | 2.50 |

## D.7 EFFECT OF TOP CANDIDATES

Table 13: Comparison of in-silico fitness evaluation for baselines and proposed method on medium/hard difficulty splits of GFP and AAV datasets. We report average (standard deviation) of 5 different runs.

| Method | AAV (hard) | | | | AAV (medium) | | | |
|---|---|---|---|---|---|---|---|---|
| | Fitness$_{10}$ ↑ | Fitness$_{100}$ ↑ | Fitness$_{1000}$ ↑ | Fitness$_{all}$ ↑ | Fitness$_{10}$ ↑ | Fitness$_{100}$ ↑ | Fitness$_{1000}$ ↑ | Fitness$_{all}$ ↑ |
| Sampling | 9.33(0.43) | 7.42(0.30) | 5.32(0.20) | 3.32(0.07) | 9.33(0.43) | 7.42(0.30) | 5.32(0.20) | 3.32(0.07) |
| ICE | 9.19 (0.11) | 8.18(0.01) | 6.67(0.01) | 3.83(0.01) | 10.71(0.05) | 9.43(0.04) | 7.64(0.03) | 4.42(0.03) |
| Align-plm | 9.91(0.00) | 9.22(0.00) | 7.90(0.00) | 5.35(0.00) | 9.54(0.00) | 8.70(0.00) | 7.33(0.00) | 4.99(0.00) |
| EXO | **12.63(0.47)** | **10.95(0.17)** | **9.41(0.08)** | **6.54(0.04)** | **13.32(0.42)** | **13.18(0.33)** | **12.46(0.14)** | **9.87(0.13)** |

| Method | GFP (hard) | | | | GFP (medium) | | | |
|---|---|---|---|---|---|---|---|---|
| | Fitness$_{10}$ ↑ | Fitness$_{100}$ ↑ | Fitness$_{1000}$ ↑ | Fitness$_{all}$ ↑ | Fitness$_{10}$ ↑ | Fitness$_{100}$ ↑ | Fitness$_{1000}$ ↑ | Fitness$_{all}$ ↑ |
| Sampling | 2.43(0.13) | 1.94(0.09) | 1.28(0.12) | 1.22(0.01) | 2.43(0.13) | 1.94(0.09) | 1.28(0.12) | 1.22(0.01) |
| ICE | 2.22(0.02) | 2.07(0.01) | 1.61(0.01) | 1.32(0.01) | 2.94(0.10) | 2.39(0.03) | 1.72(0.01) | 1.45(0.01) |
| Align-plm | 2.86(0.00) | 2.39(0.00) | 1.60(0.00) | 1.33(0.00) | 2.27(0.00) | 2.12(0.00) | 1.87(0.00) | 1.60(0.00) |
| EXO | **3.74(0.07)** | **2.81(0.07)** | **1.70(0.02)** | **1.40(0.02)** | **4.09(0.01)** | **4.04(0.01)** | **3.85(0.01)** | **3.58(0.01)** |

# E REPRODUCIBILITY

## E.1 WITHOUT SCORER IN INFERENCE

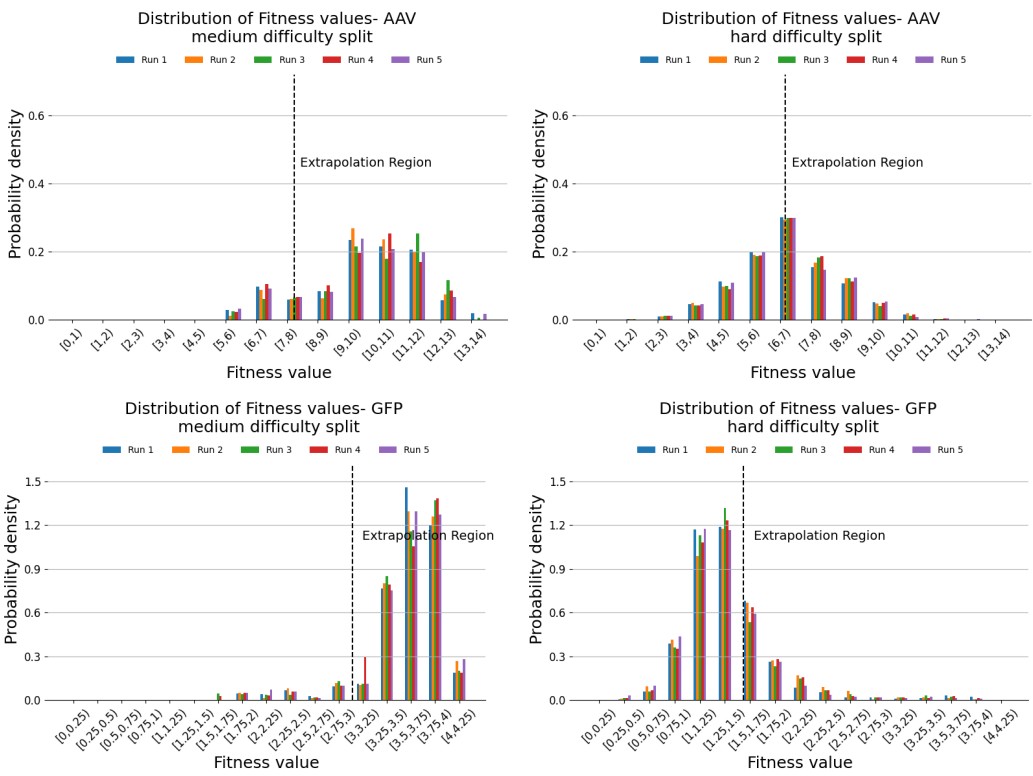

Figure 13: In-silico fitness evaluation for EXO algorithm on medium/hard difficulty splits of GFP and AAV datasets with 5 different random seed.

Table 14: In-silico fitness evaluation for EXO algorithm on medium/hard difficulty splits of GFP and AAV datasets with 5 different random seeds.

| Method | AAV (hard) | | | | AAV (medium) | | | |
|---|---|---|---|---|---|---|---|---|
| | Extrapolation ↑ | Fitness$_{100}$ ↑ | Distance$_{100}$ ↓ | Diversity$_{100}$ | Extrapolation ↑ | Fitness$_{100}$ ↑ | Distance$_{100}$ ↓ | Diversity$_{100}$ |
| Run 1 | 50.41 | 11.12 | 5.69 | 8.60 | 83.66 | 13.50 | 1.30 | 0.84 |
| Run 2 | 54.58 | 11.19 | 5.58 | 8.65 | 83.71 | 12.80 | 2.00 | 0.75 |
| Run 3 | 54.79 | 10.83 | 5.53 | 8.77 | 87.94 | 13.32 | 1.90 | 2.00 |
| Run 4 | 52.7 | 10.85 | 5.11 | 8.05 | 86.22 | 12.76 | 2.00 | 0.93 |
| Run 5 | 51.27 | 10.76 | 4.61 | 8.24 | 83.83 | 13.50 | 1.00 | 0.90 |
| AVG | 52.75 | 10.95 | 5.30 | 8.46 | 85.07 | 13.18 | 1.64 | 1.08 |
| STD | 1.74 | 0.17 | 0.39 | 0.27 | 1.72 | 0.33 | 0.41 | 0.46 |
| Method | GFP (hard) | | | | GFP (medium) | | | |
| | Extrapolation ↑ | Fitness$_{100}$ ↑ | Distance$_{100}$ ↓ | Diversity$_{100}$ | Extrapolation ↑ | Fitness$_{100}$ ↑ | Distance$_{100}$ ↓ | Diversity$_{100}$ |
| Run 1 | 22.40 | 2.70 | 6.69 | 10.36 | 92.65 | 4.06 | 2.20 | 2.52 |
| Run 2 | 25.70 | 2.86 | 5.96 | 9.47 | 92.95 | 4.04 | 2.08 | 2.56 |
| Run 3 | 23.25 | 2.90 | 16.63 | 29.09 | 92.30 | 4.04 | 2.09 | 2.84 |
| Run 4 | 26.85 | 2.75 | 10.01 | 15.97 | 93.10 | 4.05 | 2.36 | 2.42 |
| Run 5 | 23.15 | 2.86 | 6.14 | 9.92 | 93.10 | 4.04 | 1.93 | 2.50 |
| AVG | 24.27 | 2.81 | 9.08 | 14.96 | 92.82 | 4.04 | 2.13 | 2.56 |
| STD | 1.70 | 0.07 | 4.04 | 7.44 | 0.30 | 0.01 | 0.14 | 0.14 |

## E.2 EFFECT OF SCORER

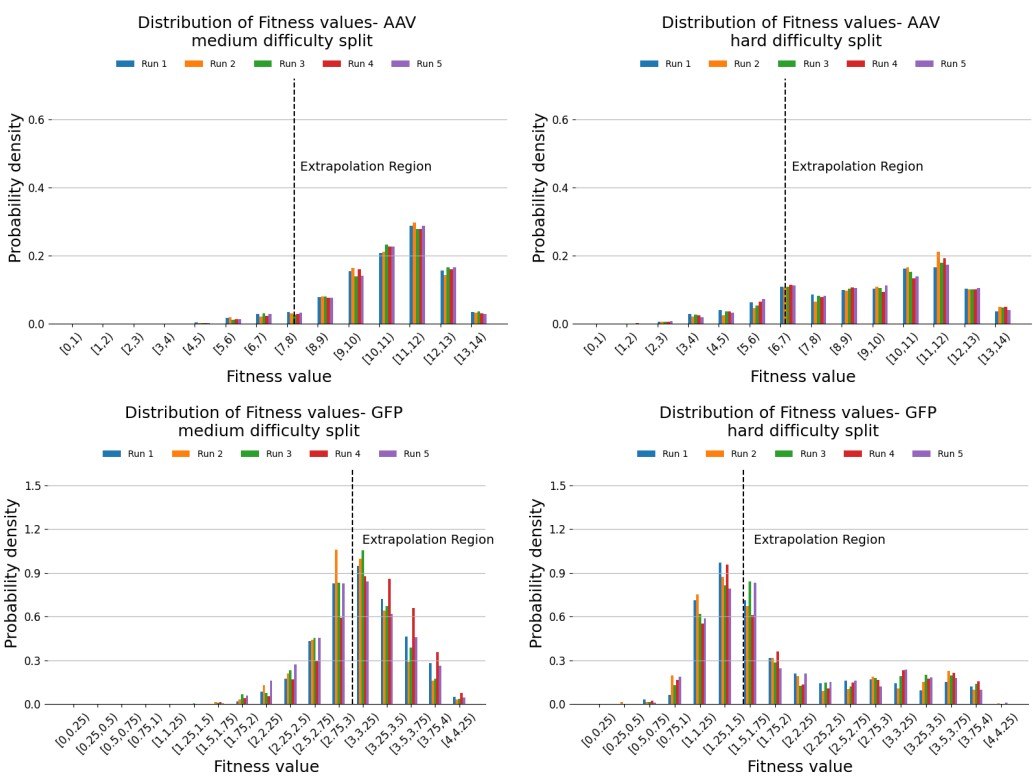

Figure 14: In-silico fitness evaluation for EXO algorithm with scorer on medium/hard difficulty splits of GFP and AAV datasets with 5 different random seed.

Table 15: In-silico fitness evaluation for EXO algorithm with scorer on medium/hard difficulty splits of GFP and AAV datasets with 5 different random seeds.

| Method | AAV (hard) | | | | AAV (medium) | | | |
|---|---|---|---|---|---|---|---|---|
| | Extrapolation $\uparrow$ | Fitness$_{100}$ $\uparrow$ | Distance$_{100}$ $\downarrow$ | Diversity$_{100}$ | Extrapolation $\uparrow$ | Fitness$_{100}$ $\uparrow$ | Distance$_{100}$ $\downarrow$ | Diversity$_{100}$ |
| Run 1 | 83.25% | 14.20 | 1.84 | 3.10 | 93.97% | 13.83 | 2.09 | 3.50 |
| Run 2 | 83.59% | 14.46 | 1.29 | 2.48 | 95.14% | 13.96 | 1.95 | 2.62 |
| Run 3 | 84.10% | 14.25 | 1.35 | 2.58 | 94.53% | 13.92 | 1.91 | 2.88 |
| Run 4 | 86.22% | 14.23 | 1.72 | 2.28 | 94.19% | 13.92 | 2.00 | 3.17 |
| Run 5 | 83.04% | 14.13 | 1.42 | 2.19 | 93.32% | 13.85 | 2.06 | 3.07 |
| AVG | 84.04% | 14.25 | 1.52 | 2.52 | 94.23% | 13.89 | 2.00 | 3.04 |
| STD | 1.14% | 0.11 | 0.21 | 0.31 | 0.60% | 0.05 | 0.06 | 0.29 |

| Method | GFP (hard) | | | | GFP (medium) | | | |
|---|---|---|---|---|---|---|---|---|
| | Extrapolation $\uparrow$ | Fitness$_{100}$ $\uparrow$ | Distance$_{100}$ $\downarrow$ | Diversity$_{100}$ | Extrapolation $\uparrow$ | Fitness$_{100}$ $\uparrow$ | Distance$_{100}$ $\downarrow$ | Diversity$_{100}$ |
| Run 1 | 53.00% | 3.78 | 1.78 | 3.10 | 53.94% | 3.96 | 3.07 | 4.18 |
| Run 2 | 52.84% | 3.81 | 1.50 | 2.65 | 69.39% | 4.00 | 2.63 | 3.92 |
| Run 3 | 54.10% | 3.82 | 1.72 | 3.16 | 56.69% | 3.92 | 2.69 | 4.05 |
| Run 4 | 45.60% | 3.78 | 1.86 | 3.31 | 50.84% | 3.90 | 2.64 | 4.13 |
| Run 5 | 49.00% | 3.77 | 1.81 | 3.18 | 59.59% | 3.96 | 2.71 | 3.91 |
| AVG | 50.90% | 3.79 | 1.73 | 3.08 | 58.09% | 3.94 | 2.74 | 4.03 |
| STD | 3.16% | 0.02 | 0.12 | 0.22 | 6.35% | 0.03 | 0.16 | 0.10 |

# F  HYPER-PARAMETER TUNING FOR PROPOSED METHOD

## F.1  PREFERENCE LEARNING PARAMETER: $\beta$

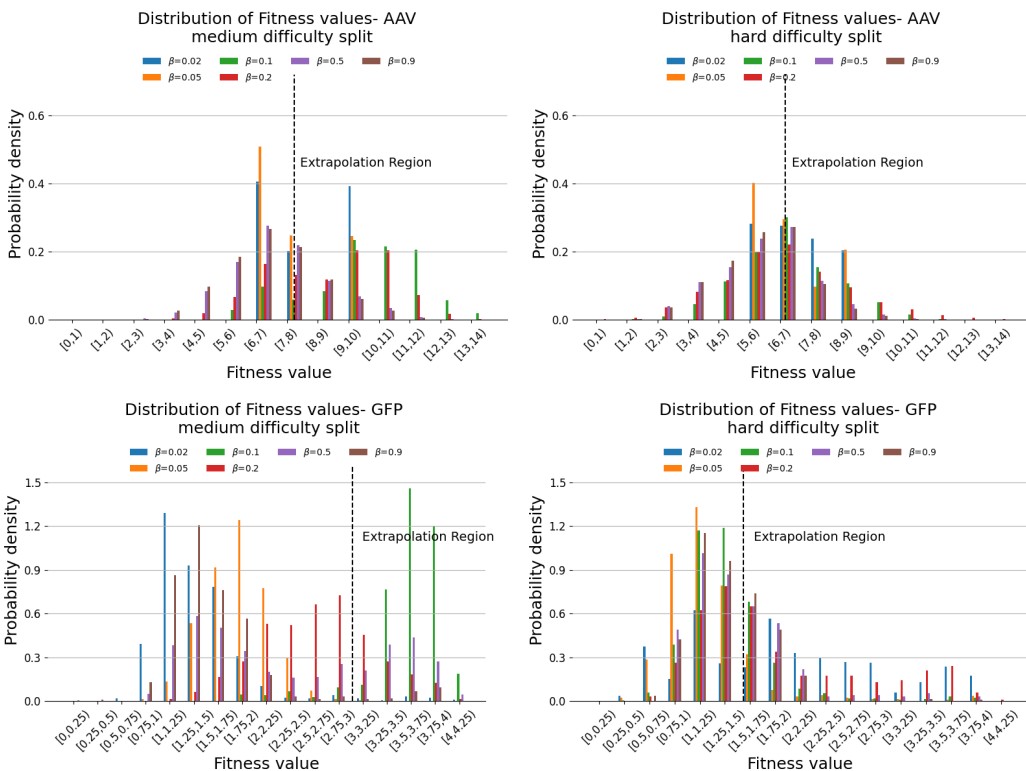

Figure 15: In-silico fitness evaluation for EXO algorithm with scorer on medium/hard difficulty splits of GFP and AAV datasets with different $beta$ hyper-parameters.

Table 16: In-silico fitness evaluation for EXO algorithm on medium/hard difficulty splits of GFP and AAV datasets with different $\beta$ hyper-parameters.

| Method | AAV (hard) | | | | AAV (medium) | | | |
|---|---|---|---|---|---|---|---|---|
| | Extrapolation $\uparrow$ | Fitness$_{100}$ $\uparrow$ | Distance$_{100}$ $\downarrow$ | Diversity$_{100}$ | Extrapolation $\uparrow$ | Fitness$_{100}$ $\uparrow$ | Distance$_{100}$ $\downarrow$ | Diversity$_{100}$ |
| $\beta$=0.02 | 64.14% | 8.57 | 8.00 | 0.00 | 39.20% | 9.85 | 7.00 | 0.00 |
| $\beta$=0.05 | 59.84% | 8.32 | 9.99 | 0.00 | 24.50% | 9.39 | 7.00 | 0.0 |
| $\beta$=0.1 | 51.27% | 10.76 | 4.61 | 8.24 | 83.83% | 13.50 | 1.00 | 0.99 |
| $\beta$=0.2 | 47.06% | 12.54 | 3.64 | 6.41 | 67.29% | 12.77 | 2.95 | 5.03 |
| $\beta$=0.5 | 33.20% | 10.25 | 6.07 | 9.35 | 32.19% | 11.56 | 5.25 | 7.80 |
| $\beta$=0.9 | 29.95% | 9.66 | 7.38 | 10.95 | 29.08% | 11.25 | 6.13 | 8.01 |

| Method | GFP (hard) | | | | GFP (medium) | | | |
|---|---|---|---|---|---|---|---|---|
| | Extrapolation $\uparrow$ | Fitness$_{100}$ $\uparrow$ | Distance$_{100}$ $\downarrow$ | Diversity$_{100}$ | Extrapolation $\uparrow$ | Fitness$_{100}$ $\uparrow$ | Distance$_{100}$ $\downarrow$ | Diversity$_{100}$ |
| $\beta$=0.02 | 62.55% | 3.87 | 0.88 | 1.17 | 1.84% | 2.99 | 2.78 | 4.82 |
| $\beta$=0.05 | 11.40% | 2.68 | 14.16 | 24.65 | 0.10% | 2.55 | 5.85 | 6.22 |
| $\beta$=0.1 | 23.15% | 2.86 | 6.14 | 9.92 | 93.10% | 4.04 | 1.93 | 2.50 |
| $\beta$=0.2 | 51.60% | 3.77 | 1.98 | 3.56 | 25.50% | 3.82 | 2.22 | 3.84 |
| $\beta$=0.5 | 37.25% | 3.13 | 2.72 | 4.20 | 33.20% | 3.94 | 1.96 | 3.00 |
| $\beta$=0.9 | 32.05% | 2.28 | 9.24 | 12.00 | 4.95% | 3.68 | 2.20 | 3.91 |

## F.2 SAMPLING PARAMETER: TEMPERATURE ($\tau$)

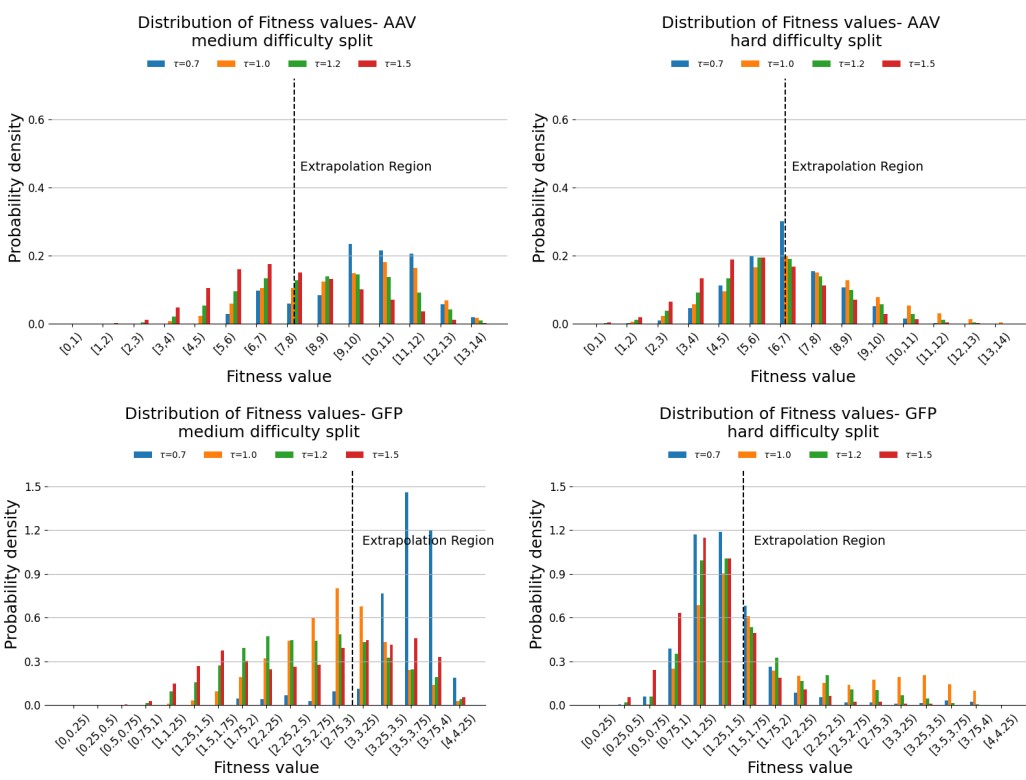

Figure 16: In-silico fitness evaluation for EXO algorithm with scorer on medium/hard difficulty splits of GFP and AAV datasets with different sampling temperature ($\tau$) hyper-parameters.

Table 17: In-silico fitness evaluation for EXO algorithm on medium/hard difficulty splits of GFP and AAV datasets with different sampling temperature ($\tau$) hyper-parameters.

| Method | AAV (hard) | | | | AAV (medium) | | | |
|---|---|---|---|---|---|---|---|---|
| | Extrapolation ↑ | Fitness$_{100}$ ↑ | Distance$_{100}$ ↓ | Diversity$_{100}$ | Extrapolation ↑ | Fitness$_{100}$ ↑ | Distance$_{100}$ ↓ | Diversity$_{100}$ |
| $\tau$=0.7 | 51.27% | 10.76 | 4.61 | 8.24 | 83.83% | 13.50 | 1.00 | 0.99 |
| $\tau$=1.0 | 57.4% | 13.14 | 3.16 | 5.54 | 76.03% | 13.75 | 2.78 | 4.28 |
| $\tau$=1.2 | 44.73% | 12.00 | 4.33 | 7.80 | 62.94% | 13.63 | 3.01 | 4.93 |
| $\tau$=1.5 | 31.71% | 11.21 | 5.85 | 9.84 | 42.69% | 12.81 | 3.66 | 5.95 |
| Method | GFP (hard) | | | | GFP (medium) | | | |
| | Extrapolation ↑ | Fitness$_{100}$ ↑ | Distance$_{100}$ ↓ | Diversity$_{100}$ | Extrapolation ↑ | Fitness$_{100}$ ↑ | Distance$_{100}$ ↓ | Diversity$_{100}$ |
| $\tau$=0.7 | 23.15% | 2.86 | 6.14 | 9.92 | 93.10% | 4.04 | 1.93 | 2.50 |
| $\tau$=1.0 | 49.25% | 3.76 | 1.88 | 3.26 | 36.19% | 3.86 | 2.90 | 4.57 |
| $\tau$=1.2 | 35.00% | 3.15 | 3.67 | 6.59 | 29.84% | 3.92 | 2.34 | 4.10 |
| $\tau$=1.5 | 19.30% | 2.45 | 12.77 | 24.18 | 41.80% | 3.96 | 2.25 | 4.18 |

# G HYPER-PARAMETER TUNING FOR BASELINE MODELS

## G.1 WITHOUT SCORER

## G.2 SAMPLING

We have utilized Prot-T5-XL (Elnaggar et al., 2021) as a pre-trained generative baseline model. We used the same combination of top-k and top-p parameters ($k = 10$, $p = 0.95$) as the proposed method since we have not done any tuning for these parameters in the proposed method as well. Similar to the proposed method, we have performed hyper-parameter tuning for temperature ($\tau$) with the same grid search of $= [0.7, 1.0, 1.2, 1.5]$. Table 18 suggests that $\tau = 0.7$ is performing slightly better than others however within the margin of standard deviations.

Table 18: In-silico fitness evaluation for Prot-T5-XL algorithm on medium/hard difficulty splits of GFP and AAV datasets with different sampling temperature ($\tau$) hyper-parameter.

| Method | AAV (hard) | | | | AAV (medium) | | | |
|---|---|---|---|---|---|---|---|---|
| | Extrapolation ↑ | Fitness$_{100}$ ↑ | Distance$_{100}$ ↓ | Diversity$_{100}$ | Extrapolation ↑ | Fitness$_{100}$ ↑ | Distance$_{100}$ ↓ | Diversity$_{100}$ |
| $\tau$=0.7 (default) | **1.64%(0.31)** | 7.42(0.30) | 4.49(0.35) | 7.18(0.61) | **1.64%(0.31)** | 7.42(0.30) | 4.49(0.35) | 7.18(0.61) |
| $\tau$=1.0 | 1.62%(0.12) | **7.45(0.14)** | **4.42(0.10)** | 7.24(0.12) | 1.62%(0.12) | **7.45(0.14)** | **4.42(0.10)** | 7.24(0.12) |
| $\tau$=1.2 | 1.56%(0.18) | 7.31(0.12) | 4.59(0.23) | 7.49(0.25) | 1.56%(0.18) | 7.31(0.12) | 4.59(0.23) | 7.49(0.25) |
| $\tau$=1.5 | 1.60%(0.12) | 7.44(0.07) | 4.43(0.05) | 7.24(0.17) | 1.60%(0.12) | 7.44(0.07) | 4.43(0.05) | 7.24(0.17) |
| Method | GFP (hard) | | | | GFP (medium) | | | |
| | Extrapolation ↑ | Fitness$_{100}$ ↑ | Distance$_{100}$ ↓ | Diversity$_{100}$ | Extrapolation ↑ | Fitness$_{100}$ ↑ | Distance$_{100}$ ↓ | Diversity$_{100}$ |
| $\tau$=0.7 (default) | **18.52%(0.80)** | **1.94(0.09)** | **10.21(0.50)** | 20.10(1.01) | **18.52%(0.80)** | **1.94(0.09)** | **10.21(0.50)** | 20.10(1.01) |
| $\tau$=1.0 | 17.74%(0.85) | 1.87(0.02) | 10.61(0.22) | 20.87(0.46) | 17.74%(0.85) | 1.87(0.02) | 10.61(0.22) | 20.87(0.46) |
| $\tau$=1.2 | 17.67%(1.27) | 1.87(0.01) | 10.58(0.25) | 20.85(0.53) | 17.67%(1.27) | 1.87(0.01) | 10.58(0.25) | 20.85(0.53) |
| $\tau$=1.5 | 18.28%(0.39) | 1.89(0.01) | 10.54(0.17) | 20.70(0.33) | 18.28%(0.39) | 1.89(0.01) | 10.54(0.17) | 20.70(0.33) |

### G.2.1 ICE

Iterative Controlled Extrapolation (ICE) Padmakumar et al. (2023) utilizes ranked pairs of sequences to learn a local editor. We used the same combination of top-k and top-p parameters ($k = 10$, $p = 0.95$) as the proposed method since we have not done any tuning for these parameters in the proposed method as well. Similar to the proposed method, we have performed hyper-parameter tuning for temperature ($\tau$) with the same grid search of $= [0.7, 1.0, 1.2, 1.5]$. Table 19 suggests that $\tau = 0.7$ is performing slightly better than others however within the margin of standard deviations.

Table 19: In-silico fitness evaluation for ICE algorithm on medium/hard difficulty splits of GFP and AAV datasets with different sampling temperature ($\tau$) hyper-parameter.

| Method | AAV (hard) | | | | AAV (medium) | | | |
|---|---|---|---|---|---|---|---|---|
| | Extrapolation ↑ | Fitness$_{100}$ ↑ | Distance$_{100}$ ↓ | Diversity$_{100}$ | Extrapolation ↑ | Fitness$_{100}$ ↑ | Distance$_{100}$ ↓ | Diversity$_{100}$ |
| $\tau$=0.7 (default) | **5.58%(0.04)** | 8.18(0.01) | **9.08(0.14)** | 13.56(0.18) | **4.59%(0.15)** | 9.43(0.04) | 7.72(0.08) | 11.49(0.24) |
| $\tau$=1.0 | 4.36%(0.13) | 8.10(0.07) | 9.42(0.12) | 14.30(0.34) | 3.72%(0.10) | 9.10(0.07) | 8.15(0.14) | 12.93(0.26) |
| $\tau$=1.2 | 4.40%(0.08) | 8.06(0.06) | 9.47(0.23) | 14.41(0.14) | 3.84%(0.13) | 9.21(0.03) | 8.06(0.12) | 12.74(0.23) |
| $\tau$=1.5 | 4.58%(0.07) | **8.24(0.06)** | 9.19(0.12) | 14.46(0.22) | 3.97%(0.20) | 9.16(0.07) | 7.95(0.15) | 12.53(0.29) |
| Method | GFP (hard) | | | | GFP (medium) | | | |
| | Extrapolation ↑ | Fitness$_{100}$ ↑ | Distance$_{100}$ ↓ | Diversity$_{100}$ | Extrapolation ↑ | Fitness$_{100}$ ↑ | Distance$_{100}$ ↓ | Diversity$_{100}$ |
| $\tau$=0.7 (default) | **27.16%(0.80)** | **2.07(0.01)** | 10.93(0.18) | 15.76(0.64) | 0.16%(0.19) | **2.39(0.03)** | 8.47(0.21) | 14.41(0.87) |
| $\tau$=1.0 | 19.97%(0.77) | 2.03(0.01) | 11.34(0.16) | 16.76(0.35) | **0.32%(0.11)** | 2.34(0.04) | **8.28(0.16)** | 15.63(0.25) |
| $\tau$=1.2 | 15.38%(0.55) | 1.94(0.01) | 10.83(0.11) | 18.61(0.26) | 0.12%(0.02) | 2.35(0.01) | 8.31(0.08) | 16.14(0.18) |
| $\tau$=1.5 | 14.64%(0.96) | 1.92(0.02) | **9.90(0.10)** | 19.18(0.14) | 0.03%(0.02) | 2.38(0.02) | 8.42(0.08) | 16.52(0.19) |

### G.2.2 ALIGN-PLM

Align-plm Lee et al. (2023) proposed a Bradley- Terry (BT) model of ranked proteins with big fitness distances between ranked proteins. We have utilized their implementation to train and run their model in inference. In inference, they proposed to use all single-site mutations with respect to seed sequence as generator and keep top M (e.g. M = 10) sequences ranked by likelihood of the proposed model. At the end of each iteration, they will keep the top N (e.g. N = 10000) for being used as seed sequence for the next iteration. For inference, M is their only hyper-parameter. We have performed hyper-parameter tuning for M with the grid search of = $[5, 10, 20]$. Their model would need 10 days to finish 10 iterations on GFP dataset on a p3 GPU machine. Since align-plm approach uses all single-site-mutations as its generative model, so it is a deterministic model. Therefore, we are reporting its standard deviation as zero in the Table 20. We can observe that $M = 20$ is outperforming the default ($M = 10$), therefore, we will use this value.

Table 20: In-silico fitness evaluation for Align-plm algorithm on medium/hard difficulty splits of GFP and AAV datasets with different top M hyper-parameter.

| Method | AAV (hard) | | | | AAV (medium) | | | |
|---|---|---|---|---|---|---|---|---|
| | Extrapolation ↑ | Fitness$_{100}$ ↑ | Distance$_{100}$ ↓ | Diversity$_{100}$ | Extrapolation ↑ | Fitness$_{100}$ ↑ | Distance$_{100}$ ↓ | Diversity$_{100}$ |
| M=5 | **24.56%(0.00)** | **10.29(0.00)** | **6.98(0.00)** | 7.91(0.00) | 3.13%(0.00) | 8.81(0.00) | 7.32(0.00) | 5.92(0.00) |
| M=10 (default) | 22.49%(0.00) | 9.22(0.00) | 7.46(0.00) | 8.56(0.00) | 3.30%(0.00) | 8.70(0.00) | **6.83(0.00)** | 5.77(0.00) |
| M=20 | 20.76%(0.00) | 9.01(0.00) | 7.60(0.00) | 8.16(0.00) | **3.49%(0.00)** | **8.66(0.00)** | 7.29(0.00) | 6.22(0.00) |
| Method | GFP (hard) | | | | GFP (medium) | | | |
| | Extrapolation ↑ | Fitness$_{100}$ ↑ | Distance$_{100}$ ↓ | Diversity$_{100}$ | Extrapolation ↑ | Fitness$_{100}$ ↑ | Distance$_{100}$ ↓ | Diversity$_{100}$ |
| M=5 | 20.75%(0.00) | 2.51(0.00) | 8.64(0.00) | 5.11(0.00) | **0.10%(0.00)** | **2.23(0.00)** | **5.45(0.00)** | 4.95(0.00) |
| M=10 (default) | 20.30%(0.00) | 2.39(0.00) | **6.88(0.00)** | 5.62(0.00) | 0.00%(0.00) | 2.12(0.00) | 5.96(0.00) | 5.23(0.00) |
| M=20 | **54.45%(0.00)** | **2.55(0.00)** | 9.64(0.00) | 4.17(0.00) | 0.00%(0.00) | 2.12(0.00) | 6.13(0.00) | 5.41(0.00) |

## G.3 WITH SCORER

### G.3.1 ITERATIVE SAMPLING

We have utilized Prot-T5-XL (Elnaggar et al., 2021) in combination with scorer as a pre-trained language model guided by scorer to increase fitness. We used the same combination of top-k and top-p parameters (k = 10, p = 0.95) as the proposed method since we have not done any tuning for these parameters in the proposed method as well. Similar to the proposed method, we have performed hyper-parameter tuning for temperature ($\tau$) with the same grid search of = $[0.7, 1.0, 1.2, 1.5]$. Table 21 suggests that $\tau = 1.0$ (default) is performing similarly to other $\tau$ within the margin of standard deviations.

Table 21: In-silico fitness evaluation for Iterative sampling algorithm on medium/hard difficulty splits of GFP and AAV datasets with different sampling temperature ($\tau$) hyper-parameters.

| Method | AAV (hard) | | | | AAV (medium) | | | |
|---|---|---|---|---|---|---|---|---|
| | Extrapolation ↑ | Fitness$_{100}$ ↑ | Distance$_{100}$ ↓ | Diversity$_{100}$ | Extrapolation ↑ | Fitness$_{100}$ ↑ | Distance$_{100}$ ↓ | Diversity$_{100}$ |
| $\tau$=0.7 | 1.18%(0.10) | 7.64(0.07) | 6.37(0.16) | 10.82(0.37) | 0.47%(0.13) | 7.64(0.30) | 6.57(0.36) | 11.35(0.55) |
| $\tau$=1.0 (default) | 1.37%(0.12) | 7.67(0.25) | 6.54(0.44) | 11.28(0.70) | 0.40%(0.10) | 7.67(0.14) | 6.59(0.65) | 11.43(0.97) |
| $\tau$=1.2 | 1.42%(0.26) | 7.85(0.20) | 6.42(0.38) | 11.04(0.59) | 0.64%(0.05) | 7.96(0.11) | 6.18(0.19) | 10.74(0.44) |
| $\tau$=1.5 | 1.58%(0.21) | 7.90(0.33) | 6.43(0.13) | 10.87(0.28) | 0.61%(0.17) | 8.06(0.30) | 6.21(0.29) | 10.91(0.41) |
| Method | GFP (hard) | | | | GFP (medium) | | | |
| | Extrapolation ↑ | Fitness$_{100}$ ↑ | Distance$_{100}$ ↓ | Diversity$_{100}$ | Extrapolation ↑ | Fitness$_{100}$ ↑ | Distance$_{100}$ ↓ | Diversity$_{100}$ |
| $\tau$=0.7 | 0.03%(0.02) | 1.19(0.02) | 180.36(8.35) | 205.14(1.23) | 0.00%(0.00) | 1.19(0.02) | 187.21(10.78) | 207.74(2.32) |
| $\tau$=1.0 (default) | 0.01%(0.02) | 1.18(0.03) | 184.71(10.78) | 207.45(3.02) | 0.00%(0.00) | 1.22(0.04) | 174.52(17.62) | 201.22(6.04) |
| $\tau$=1.2 | 0.00%(0.00) | 1.16(0.03) | 178.61(8.32) | 205.99(3.93) | 0.00%(0.00) | 1.21(0.03) | 183.39(13.33) | 206.30(3.75) |
| $\tau$=1.5 | 0.01%(0.02) | 1.18(0.02) | 175.05(16.85) | 202.91(7.20) | 0.00%(0.00) | 1.20(0.02) | 182.03(13.60) | 204.96(4.87) |

### G.3.2 ICE + SCORER

Iterative Controlled Extrapolation (ICE) Padmakumar et al. (2023) with scorer has the same hyper-parameter ($\tau$) to be tuned as most of the other models. we have performed hyper-parameter tuning for temperature ($\tau$) with the same grid search of = $[0.7, 1.0, 1.2, 1.5]$. $\tau = 0.7$ is the default value used by the authors. Based on our hyper-parameter tuning reported in Table 22, we can observe that default value performs well on GFP hard split and $\tau = 1.5$ performs well on both medium and hard split of AAV dataset. Therefore, will use $\tau = 1.5$ in our main reporting.

Table 22: In-silico fitness evaluation for ICE algorithm with scorer on medium/hard difficulty splits of GFP and AAV datasets with different sampling temperature ($\tau$) hyper-parameters.

| Method | AAV (hard) | | | | AAV (medium) | | | |
|---|---|---|---|---|---|---|---|---|
| | Extrapolation $\uparrow$ | Fitness$_{100}$ $\uparrow$ | Distance$_{100}$ $\downarrow$ | Diversity$_{100}$ | Extrapolation $\uparrow$ | Fitness$_{100}$ $\uparrow$ | Distance$_{100}$ $\downarrow$ | Diversity$_{100}$ |
| $\tau$=0.7 (default) | 27.13%(3.18) | 9.24(0.19) | 8.08(0.19) | 10.47(0.87) | 18.14%(3.20) | 10.34(0.26) | 7.21(0.26) | 8.37(1.02) |
| $\tau$=1.0 | 33.69%(0.42) | 9.64(0.12) | 7.78(0.21) | 10.47(0.31) | 25.28%(1.23) | 10.50(0.05) | 6.89(0.17) | 9.33(0.31) |
| $\tau$=1.2 | 34.41%(0.74) | 9.77(0.18) | 7.94(0.13) | 11.18(0.54) | 29.94%(1.21) | 10.68(0.10) | 6.85(0.20) | 9.29(0.32) |
| $\tau$=1.5 | **37.01%(0.36)** | **10.26(0.12)** | **6.50(0.30)** | 9.90(0.47) | **33.17%(0.59)** | **10.80(0.10)** | **6.52(0.49)** | 9.89(0.55) |

| Method | GFP (hard) | | | | GFP (medium) | | | |
|---|---|---|---|---|---|---|---|---|
| | Extrapolation $\uparrow$ | Fitness$_{100}$ $\uparrow$ | Distance$_{100}$ $\downarrow$ | Diversity$_{100}$ | Extrapolation $\uparrow$ | Fitness$_{100}$ $\uparrow$ | Distance$_{100}$ $\downarrow$ | Diversity$_{100}$ |
| $\tau$=0.7 (default) | **28.78%(0.40)** | **2.09(0.01)** | 10.85(0.11) | 13.96(0.83) | 0.13%(0.19) | **2.66(0.05)** | 7.87(0.22) | 11.66(0.44) |
| $\tau$=1.0 | 20.68%(1.39) | 2.05(0.01) | 10.93(0.16) | 16.49(0.56) | **0.65%(0.10)** | 2.56(0.02) | 8.03(0.14) | 14.27(0.24) |
| $\tau$=1.2 | 14.30%(0.69) | 1.95(0.02) | 10.29(0.05) | 18.40(0.30) | 0.29%(0.23) | 2.54(0.05) | 8.13(0.10) | 15.14(0.18) |
| $\tau$=1.5 | 14.87%(0.75) | 1.96(0.04) | **9.52(0.25)** | 18.19(0.46) | 0.02% (0.02) | 2.53(0.04) | 8.22(0.14) | 15.68(0.28) |

### G.3.3 BiGGS

BiGGS Kirjner et al. (2024) proposed Gibbs sampling with Graph-based Smoothing in the smoothed fitness landscape. The main hyper-parameter, in their inference is the temperature ($\tau$) used in Gibbs sampling. Their default is $\tau = 0.01$. We have performed hyper-parameter tuning for temperature ($\tau$) with the grid search of $= [0.002, 0.005, 0.01, 0.02]$. Based on Table 23, we can observe that $\tau = 0.002$ is having serious issues with diversity. Particularly, it has diversity of zero for medium splits of AAV and GFP datasets for top 100 designs. In addition, we can observe that for hard splits of AAV and GFP, default value ($\tau = 0.01$) is performing better than ($\tau = 0.005$) while ($\tau = 0.005$) is performing better on medium splits of AAV and GFP datasets.

Table 23: In-silico fitness evaluation for BiGGS algorithm on medium/hard difficulty splits of GFP and AAV datasets with different sampling temperature ($\tau$) hyper-parameters.

| Method | AAV (hard) | | | | AAV (medium) | | | |
|---|---|---|---|---|---|---|---|---|
| | Extrapolation $\uparrow$ | Fitness$_{100}$ $\uparrow$ | Distance$_{100}$ $\downarrow$ | Diversity$_{100}$ | Extrapolation $\uparrow$ | Fitness$_{100}$ $\uparrow$ | Distance$_{100}$ $\downarrow$ | Diversity$_{100}$ |
| $\tau$=0.002 | **32.2%(3.67)** | 10.09(0.32) | 3.95(0.09) | 0.128 (0.25) | **25.98%(1.27)** | 11.56(0) | 5.99(0) | 0.00(0.00) |
| $\tau$=0.005 | 18.70%(1.94) | 9.35(0.30) | **3.20(0.25)** | 3.56(0.44) | 9.31%(1.28) | **12.37(0.19)** | 5.74(0.46) | 5.30(0.24) |
| $\tau$=0.01 (default) | 16.80%(5.37) | **10.85(0.51)** | 5.70(1.06) | 6.38(1.44) | 4.88%(0.84) | 10.21(0.88) | 8.05(0.84) | 8.34(0.93) |
| $\tau$=0.02 | 26.19%(2.40) | 10.01(0.09) | 4.28(0.31) | 3.79(0.13) | 7.07%(0.70) | 11.28(0.43) | 4.65(1.04) | 7.36(0.97) |

| Method | GFP (hard) | | | | GFP (medium) | | | |
|---|---|---|---|---|---|---|---|---|
| | Extrapolation $\uparrow$ | Fitness$_{100}$ $\uparrow$ | Distance$_{100}$ $\downarrow$ | Diversity$_{100}$ | Extrapolation $\uparrow$ | Fitness$_{100}$ $\uparrow$ | Distance$_{100}$ $\downarrow$ | Diversity$_{100}$ |
| $\tau$=0.002 | **100%(0.0)** | **3.97(0.01** | 0.66(0.39) | 0.24(0.22) | **100%(0.0)** | 3.99(0.04) | 0.40(0.48) | 0.00(0.00) |
| $\tau$=0.005 | 89.30% (3.04) | 3.80(0.02) | 1.68(0.20) | 3.21(0.39) | 98.92% (1.38) | **4.05(0.01)** | 2.21(1.15) | 1.96(0.70) |
| $\tau$=0.01 (default) | 99.53%(0.21) | 3.83(0.02) | 3.48(0.36) | 6.01(0.51) | 55.50%(6.75) | 3.89(0.03) | 4.13(0.38) | 5.74(0.71) |
| $\tau$=0.02 | 95.58%(0.97) | 3.88(0.01) | 2.36(0.15) | 4.39(0.22) | 67.91%(4.27) | 3.91(0.01) | 2.64(0.22) | 5.23(0.36) |

### G.3.4 LatprotRL

LatprotRL Lee et al. (2024) proposed protein fitness optimization through reinforcement learning in latent space of large language models. Maximum number of mutations allowed per step is the only hyper-parameter controllable in inference of LatprotRL. We have performed hyper-parameter tuning for maximum number of mutations allowed per step with the grid search of $= [2, 3, 4, 5]$. When we set maximum number of mutations allowed per step equal 2, we have encounter errors in the inference for hard splits of AAV and GFP. Based on the results reported in Table 24, the current default value (3) is already performing robustly well on these datasets.

Table 24: In-silico fitness evaluation for LatprotRL algorithm on medium/hard difficulty splits of GFP and AAV datasets with different maximum number of mutations per step hyper-parameter.

| Method | AAV (hard) | | | | AAV (medium) | | | |
|---|---|---|---|---|---|---|---|---|
| | Extrapolation $\uparrow$ | Fitness$_{100}$ $\uparrow$ | Distance$_{100}$ $\downarrow$ | Diversity$_{100}$ | Extrapolation $\uparrow$ | Fitness$_{100}$ $\uparrow$ | Distance$_{100}$ $\downarrow$ | Diversity$_{100}$ |
| max_mutations_per_step=2 | - | - | - | - | **38.96%(1.62)** | 12.47(0.09) | **2.78(0.05)** | 4.85(0.22) |
| max_mutations_per_step=3 (default) | **64.82%(1.02)** | **13.29(0.06)** | **2.45(0.16)** | 4.67(0.23) | 38.63%(0.86) | 12.53(0.08) | 2.83(0.15) | 5.21(0.07) |
| max_mutations_per_step=4 | 62.57%(0.85) | 13.22(0.11) | 2.82(0.09) | 5.10(0.11) | 37.45%(1.38) | **12.60(0.11)** | 2.98(0.07) | 5.47(0.14) |
| max_mutations_per_step=5 | 60.46%(0.71) | 13.08(0.09) | 3.15(0.10) | 5.68(0.21) | 35.76%(0.98) | 12.40(0.08) | 3.08(0.11) | 5.71(0.14) |

| Method | GFP (hard) | | | | GFP (medium) | | | |
|---|---|---|---|---|---|---|---|---|
| | Extrapolation $\uparrow$ | Fitness$_{100}$ $\uparrow$ | Distance$_{100}$ $\downarrow$ | Diversity$_{100}$ | Extrapolation $\uparrow$ | Fitness$_{100}$ $\uparrow$ | Distance$_{100}$ $\downarrow$ | Diversity$_{100}$ |
| max_mutations_per_step=2 | - | - | - | - | 35.77%(1.04) | 3.91(0.01) | 1.61(0.03) | 3.15(0.04) |
| max_mutations_per_step=3 (default) | **88.28%(1.05)** | 3.88(0.01) | 1.48(0.04) | 2.86(0.07) | 38.22%(1.99) | 3.92(0.01) | **1.56(0.05)** | 3.04(0.05) |
| max_mutations_per_step=4 | 87.81%(0.88) | **3.89(0.01)** | **1.47(0.04)** | 2.86(0.05) | **39.87%(1.72)** | **3.93(0.01)** | 1.56(0.06) | 3.08(0.05) |
| max_mutations_per_step=5 | 86.19%(0.97) | 3.88(0.01) | 1.48(0.02) | 2.87(0.06) | 38.26%(1.31) | 3.93(0.01) | 1.57(0.04) | 3.08(0.07) |

# H TRAINING CURVES FOR PREFERENCE LEARNING ALGORITHMS

## H.1 MEDIUM DIFFICULTY SPLIT OF GFP

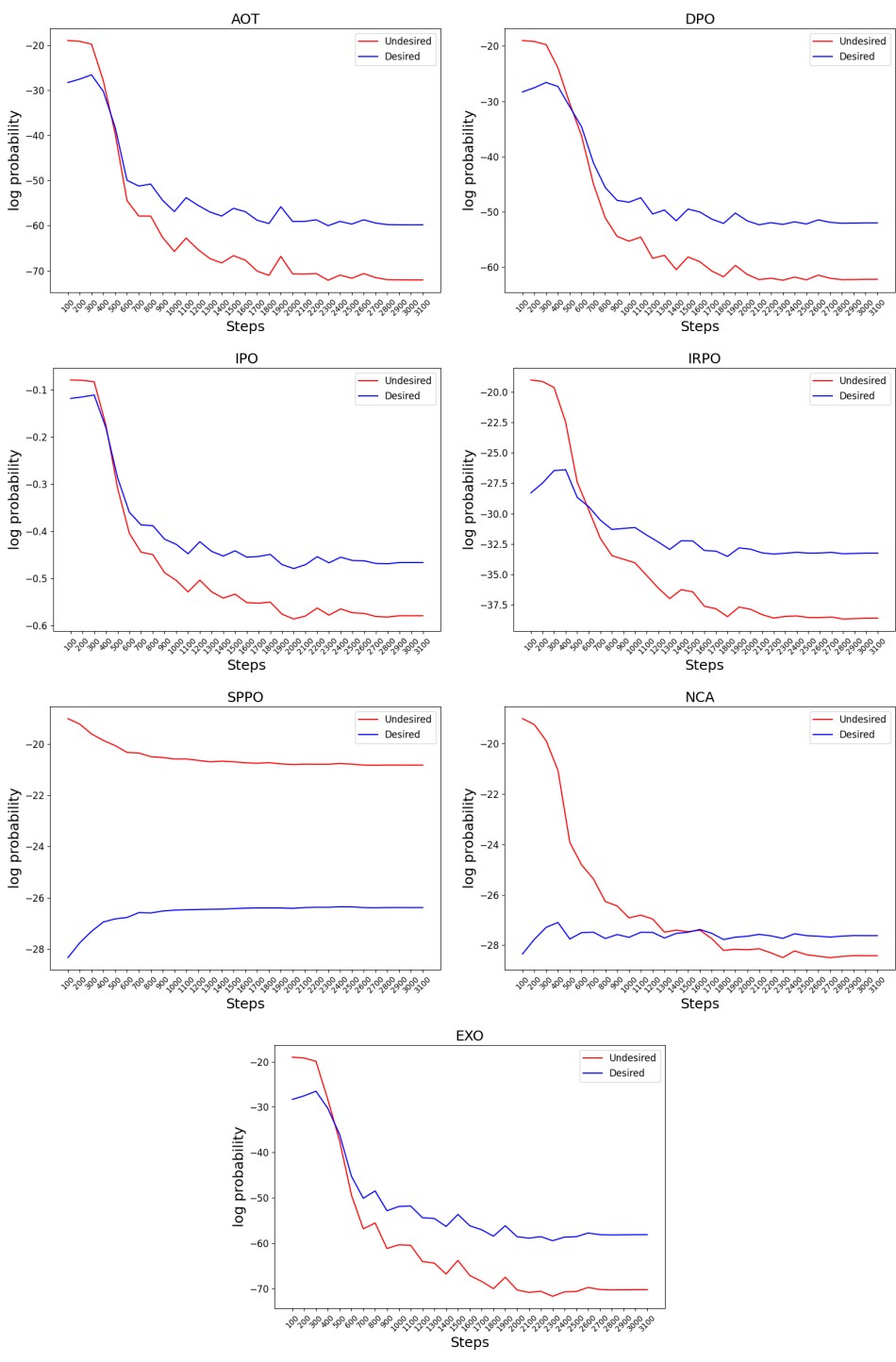

Figure 17: Log probability of validation set's desired vs undesired sequences for preference learning models on medium difficulty split of GFP.

## H.2 HARD DIFFICULTY SPLIT OF GFP

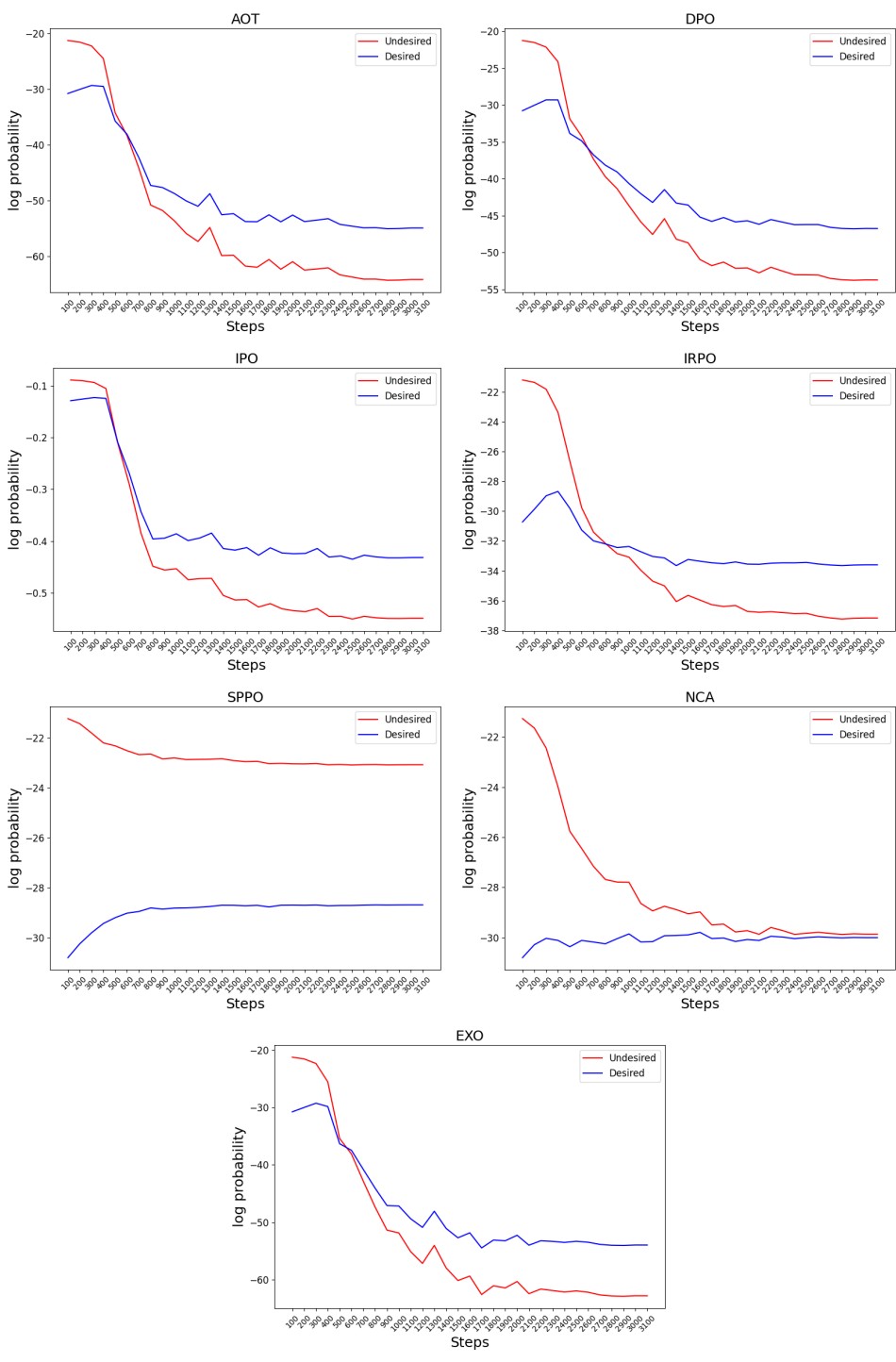

Figure 18: Log probability of validation set's desired vs undesired sequences for preference learning models on hard difficulty split of GFP.

## H.3  MEDIUM DIFFICULTY SPLIT OF AAV

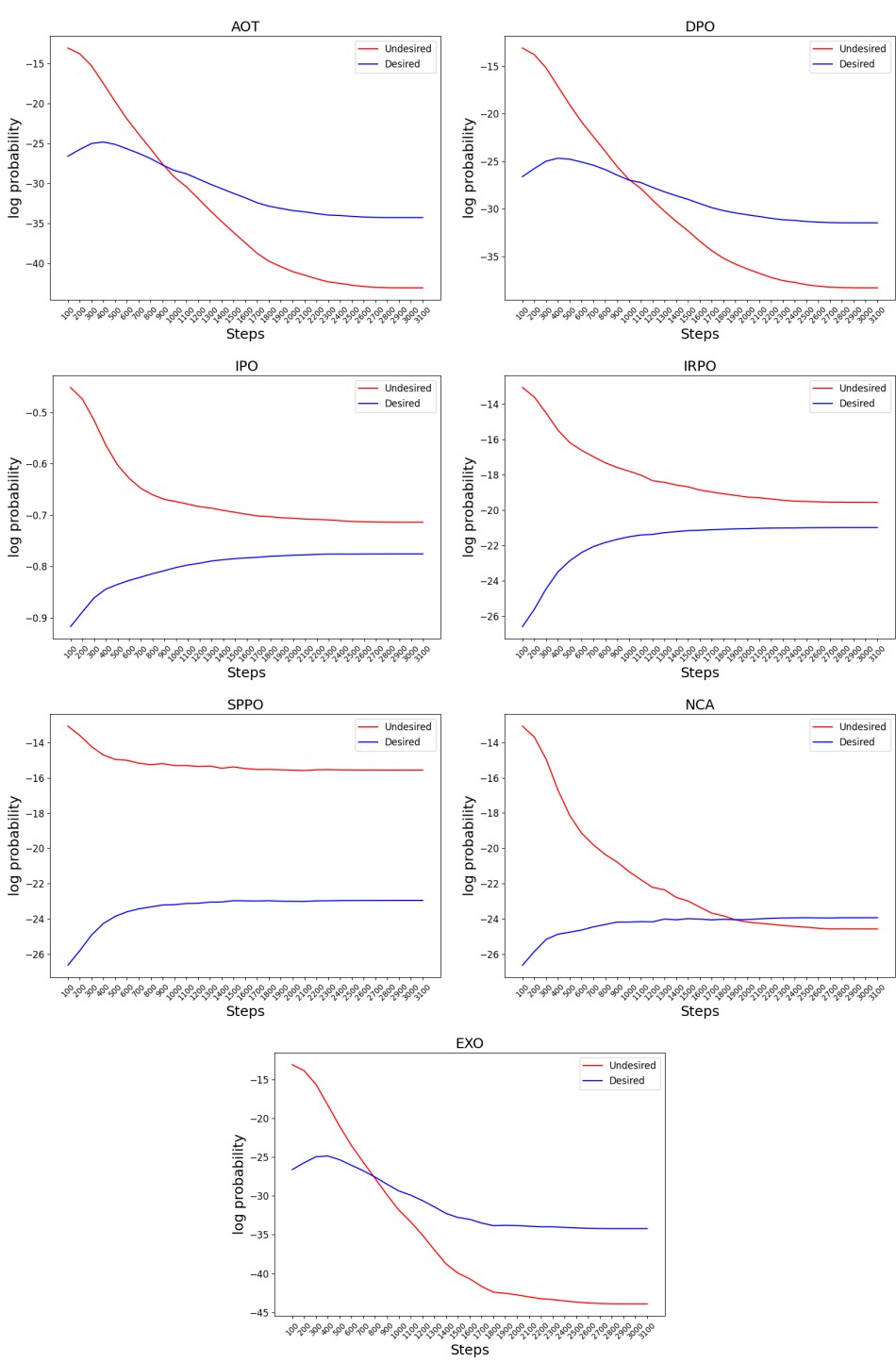

Figure 19: Log probability of validation set's desired vs undesired sequences for preference learning models on medium difficulty split of AAV.

## H.4 Hard difficulty split of AAV

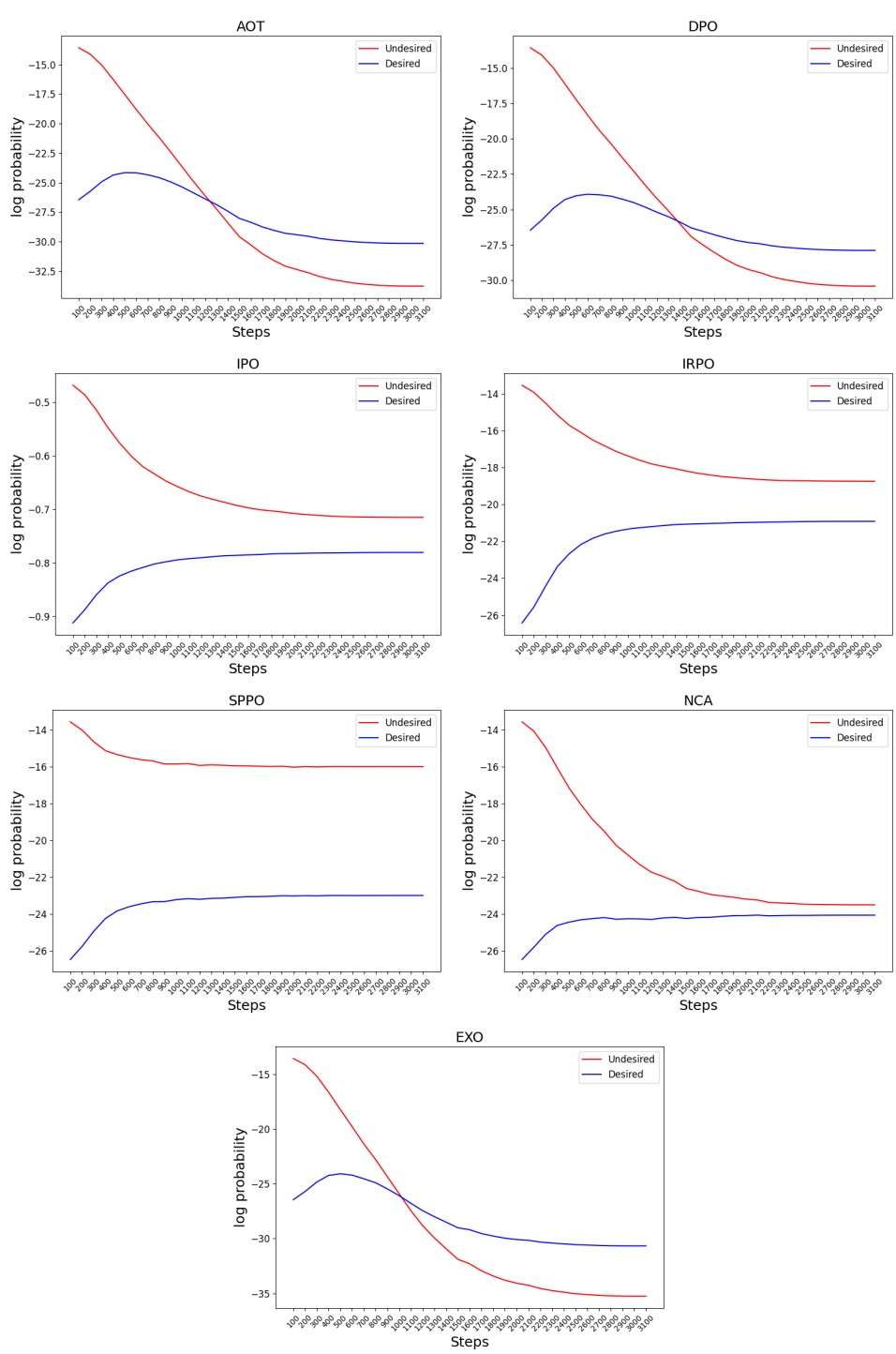

Figure 20: Log probability of validation set's desired vs undesired sequences for preference learning models on hard difficulty split of AAV.

# I   TRAINING CURVES FOR TRIPLET DATASETS

## I.1   MEDIUM DIFFICULTY SPLIT OF GFP

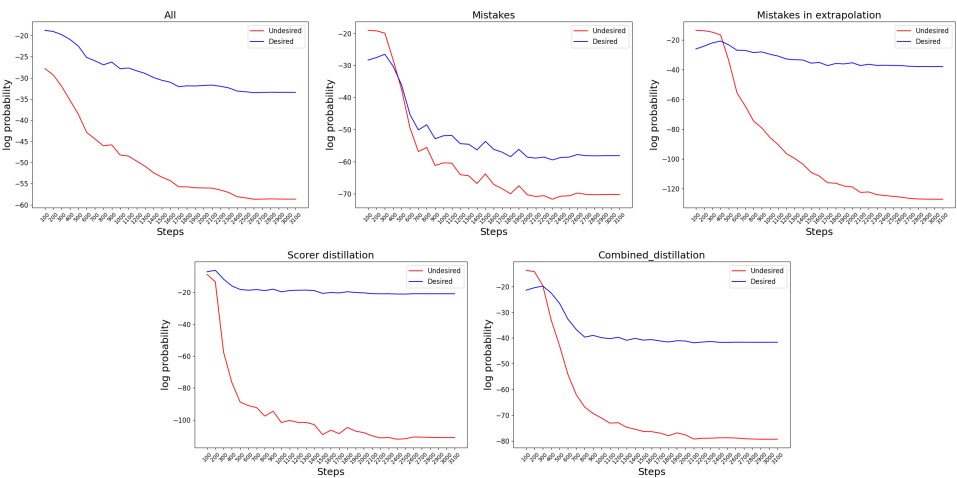

Figure 21: Log probability of validation set's desired vs undesired sequences for various approach of creating triplets on medium difficulty split of GFP.

## I.2   HARD DIFFICULTY SPLIT OF GFP

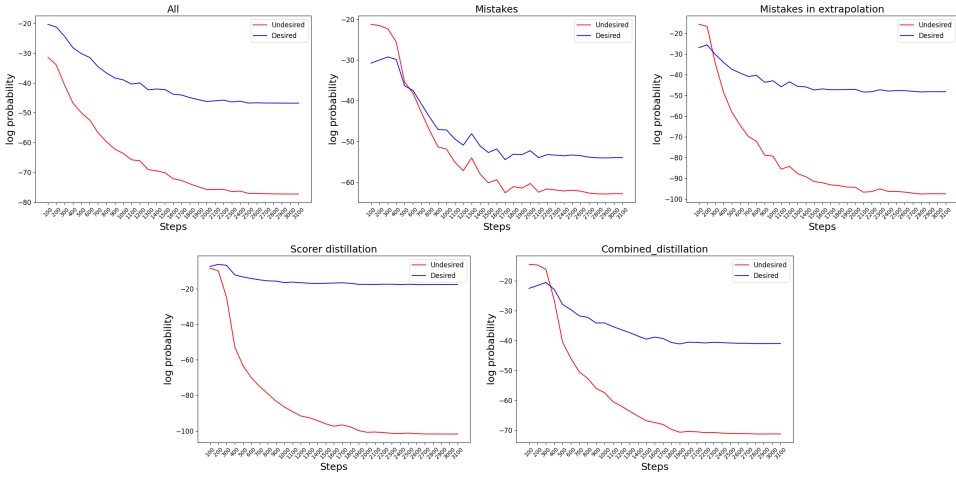

Figure 22: Log probability of validation set's desired vs undesired sequences for various approach of creating triplets on medium difficulty split of GFP.

## I.3 MEDIUM DIFFICULTY SPLIT OF AAV

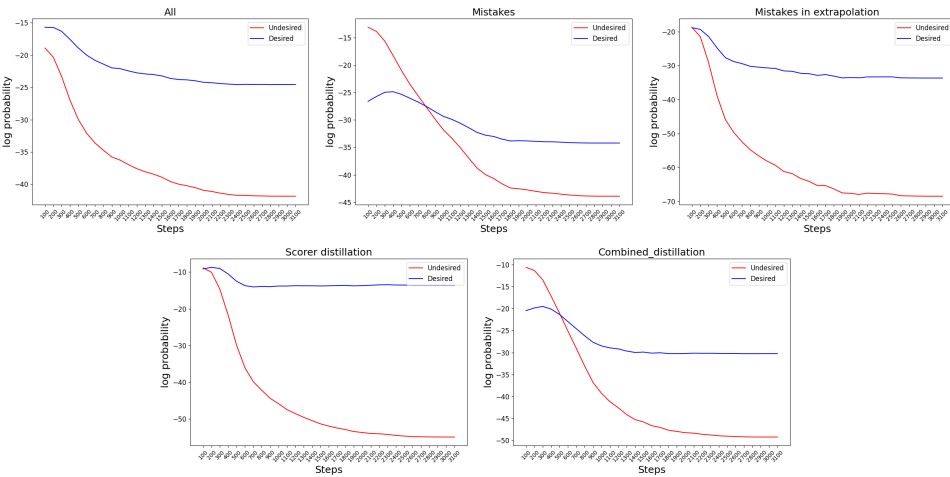

Figure 23: Log probability of validation set's desired vs undesired sequences for various approach of creating triplets on medium difficulty split of AAV.

## I.4 HARD DIFFICULTY SPLIT OF AAV

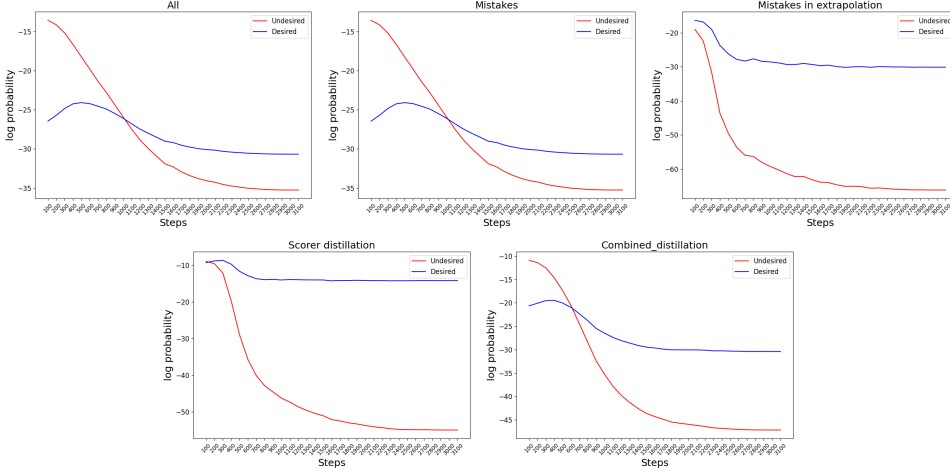

Figure 24: Log probability of validation set's desired vs undesired sequences for various approach of creating triplets on medium difficulty split of AAV.

