# OpenReview forum: "Data Distillation for extrapolative protein design through exact preference optimization"
_ICLR.cc/2025/Conference — ICLR 2025 Poster_

### Official Review · Reviewer_qbi3 · 2024-10-28

**Soundness:** 3
**Presentation:** 3
**Contribution:** 3
**Rating:** 8
**Confidence:** 3

**Summary:**

This paper addresses the problem of designing novel protein sequences with fitness values beyond those training ones. The paper distills triplet relationships into a generative model using preference alignment to better capture the fitness gradient than alternative pair-based approaches. This model is validated on AAV and GFP and outperforms state-of-the-art scorer-based and non-scorer-based baselines in extrapolation tasks.

**Strengths:**

Overall this paper is a valuable contribution to the field.
* The problem of designing proteins with fitness value beyond the training range is an important problem and challenging in protein design e.g. lead optimization; the proposed work has high practical value.
* The proposed approach is well-motivated; the use of triplet-information with preference optimization for protein design makes a lot of sense and is (to my knowledge) new. The analysis of the importance of using difficult triplets is insightful. The experimental validation is thorough (including ablation studies) and shows that the proposed model outperforms alternatives in the extrapolation regime.

**Weaknesses:**

* The preference optimization relies on a scorer function to define the triplets; in real-world setting, training such scorers is difficult and the scorer is often used in out-of-distribution settings where it does not generalize well. Adding some discussion on this topic would be valuable to the paper as it is not discussed to avoid settings where the proposed preference optimization would degrade the base pairwise model. In particular, could the authors explain how robust is the preference fine tuning model to noisy labels? How to avoid pathological cases where difficult triplets are defined by an imperfect scorer ?
* The presentation is at times confusing and difficult to understand (c.f. clarifying question).

**Questions:**

* Some parts of this paper are dense and difficult to read; in particular the definition of triplets (defined in different parts of the paper in sections 3.4; 4.3 and 6.1). W.r.t. scorer distillation, what is defined as a hard triplet ranking wise (for the vanilla version, hardness is defined by the local editor) ? The term "scorer distillation" is confusing (if I understood correctly the vanilla version also uses a scorer).
* How important is the mask infilling step ? How frequently are the mask infilled samples represented in the challenging triplets (where the editor does not capture the right fitness ordering) ?


====Post rebuttal====
I thank the authors for the rebuttal; I recommend acceptance.

---

> ### Author Response · Authors · 2024-11-25
> **Response to Reviewer qbi3**
>
> We appreciate the thoughtful comments and summary from reviewer qbi3, which have helped improve the quality of the manuscript.
>
> ### **W1: regarding preference optimization relying on a scorer function to define the triplets**
>
> We thank the reviewer for pointing out this important concern. We hope that clarifying the difference between scorer and evaluator will mitigate the concern. Scorer is a model that only sees the training region and therefore might perform poorly in out-of-distribution scenario as rightly suggested by the reviewer. Therefore, the current scorer used in our model is noisy particularly with respect to sequence in extrapolation region. On the other hand, the in-silico evaluator is a model which is trained on both training and extrapolating regions. Therefore, it is expected to perform better in the extrapolation region and can be thought as a noisy surrogate of wet-lab experiments. In section C (scorer vs evaluator), we have shown that scorer and in-silico evaluator do not correlate well in the extrapolation region as expected.
>
> ### **Q1: clarification regarding definition of hard triplet ranking wise for scorer distillation**
> We thank the reviewer for the clarification question. We have updated section 6.1 in main text to better define the hardness for scorer distillation. Particularly, for a given $x_{prompt}$, one can sample $K$ sequences $y_{1:K}=[y_1, y_2, .., y_K]$ from $P_{pair} (.|x_{prompt})$. We choose the triplewise rankings for all pairs based on $s_{i,j} = f(y_j) - f(y_i)$ where $f(y_j) < f(x_{prompt}) < f(y_i)$. By definition, $s_{ij} > 0$ can be considered as hard example since the pairwise model has wrongly generated $y_j$ with lower predicted fitness than prompt. We would like to encourage the pairwise model to generate more sequences similar to $y_i$ since its predicted fitness is higher than prompt sequence. This approach can be considered as scorer distillation since we are attempting to distill the scorer's knowledge in both training and extrapolation regions into the generative model.
>
> ### **Q2: re how important is the mask infilling step**
> We thank the reviewer for asking this detailed question. The mask infilling step is important while training the reference pairwise model. We followed the recommendation from [1] to do masking-infilling starting from the training sequences (seeds). Scorer function is utilized to assess whether the newly generated pair (seed, sequence) has a small but meaningful improvement in the desired direction. Since the focus of the paper was on triplet creation rather than pairs, we have not done ablation studies on importance of mask infilling for pairwise models. We followed the recommendation of [1].
>
> [1] Padmakumar, Vishakh, et al. "Extrapolative controlled sequence generation via iterative refinement." ICML 2023.

---

> > ### Author Response · Authors · 2024-12-01
> > **Further discussion**
> >
> > We really appreciate the reviewer for helping us improving the manuscript quality. We would like to ask if our response has addressed your concerns? If not, we are very happy to discuss and further clarify.
> >
> > Kind Regards, Authors

---

> > > ### Comment · Reviewer_qbi3 · 2024-12-01
> > >
> > > I thank the authors for the rebuttal; I increased my score.

---

### Official Review · Reviewer_jZ4t · 2024-11-03

**Soundness:** 3
**Presentation:** 2
**Contribution:** 2
**Rating:** 6
**Confidence:** 3

**Summary:**

The paper introduces a data distillation method focused on extrapolative protein design by applying triplet-based preference optimization. The approach, termed Exact Preference Optimization (EXO), aims to improve the generation of protein sequences with fitness values that exceed those observed during training. By aligning model training with structured triplet preferences, the authors seek to direct the model toward higher-fitness extrapolation regions. Performance is evaluated on Green Fluorescent Proteins (GFP) and Adeno-Associated Virus (AAV) datasets, with results suggesting improvements over existing methods in metrics such as extrapolation rate and top-100 fitness.

**Strengths:**

1. The method provides a unique approach to data distillation for extrapolative tasks, using triplet preferences instead of traditional pairwise comparisons. This formulation may better approximate the gradient direction toward higher fitness values, which is often challenging in extrapolative settings.

2. The paper includes benchmark results against several existing methods, both scorer-based and non-scorer-based, which helps to establish a comparative baseline for their approach. The presented improvements in extrapolation metrics provide an indication that the model is capturing some aspects of fitness not fully addressed by baseline methods.

3. Evaluation across multiple dimensions, including extrapolation rate and diversity among top sequences, gives a broader perspective on how the model navigates the trade-off between exploration and exploitation. This helps to contextualize the method’s potential in extrapolative design, though the gains remain somewhat dataset-specific.

4. Ablation studies on different strategies for preference data construction are included, offering insights into the model’s sensitivity to triplet selection and shedding light on the specific design choices that contribute to observed performance improvements.

**Weaknesses:**

1. Generating triplet preferences, particularly when using scorer-based distillation, is computationally intensive and may limit the scalability of this method in practice. The reliance on costly preference data creation steps could hinder its application to larger datasets or more varied protein types.

2. The paper’s evaluation is confined to GFP and AAV datasets. While they are common in the field, they may not fully represent the diversity of protein design challenges. The extent to which the method generalizes across other types of proteins remains unclear, and additional benchmark results would strengthen claims.

3. The approach relies heavily on in-silico evaluators that are trained on subsets of the data, which could contain biases and limit the evaluator’s ability to accurately assess extrapolative performance. The model’s extrapolative capability, therefore, may not fully translate to real-world scenarios without further validation, such as wet-lab testing. This dependence on in-silico metrics calls into question the practical value of the observed improvements.

**Questions:**

1. The triplet generation process, especially when involving scorer-based distillation, appears computationally intensive. Could the authors provide more detail on the computational requirements for this step? Additionally, how would the method scale to larger datasets or to proteins with significantly different characteristics?

2. The results are presented for GFP and AAV datasets. To what extent do authors believe the method could generalize to other proteins, especially those with less predictable fitness landscapes? Have the authors considered additional datasets, or are there plans to test the model on other types of proteins?

3. The ablation studies show some performance sensitivity to triplet dataset construction. Could the authors elaborate on the specific criteria used to select the triplets, and whether the approach is robust to different triplet selection methods? Additionally, how dependent is the performance on the accuracy of the scorer used in the distillation process?

4. The diversity metric suggests some degree of balance between exploration and exploitation, but it’s unclear how this affects overall model performance. Could the author clarify how the model’s exploration-exploitation balance was tuned or controlled, and how this may impact its ability to generalize in truly extrapolative regions?

---

> ### Author Response · Authors · 2024-11-25
> **Response to Reviewer jZ4t**
>
> We appreciate the thoughtful comments and summary from reviewer jZ4t, which have helped improve the quality of the manuscript.
>
> ### **W1-Q1: clarification question regarding the computational cost of scorer-distillation**
> We thank reviewer for highlighting this important concern. The reviewer has correctly pointed out that scorer-distillation is computationally expensive and that is the main reason we have not used this approach as our default. We have added table 7 in supplementary for computational cost of creating one triplet based on scorer distillation or the default one. We added a few lines in section 6.1 in main text to highlights this. As mentioned in the revised paper, scorer distillation is more than 200 times more expensive to run on p3 GPU (V100) machines in comparison to the default for GFP dataset (280 length protein).
>
> ### **W2-Q2: Regarding benchmarking the proposed methods against all baseline on a new dataset**
> We thank the reviewer for their excellent recommendation to further benchmarking on new datasets. While it is always better to show robustness of the proposed method on more datasets, due to the time limitation, we were not able to add another dataset. However, we hope that additional experimentations (consecutive vs non-consecutive bins, sequence similarity constrains) and thorough hyper-parameter tuning for all baselines help in showing the robustness of the proposed approach.
>
> ### **W3: evaluation through in-sillico evaluators that are trained on whole data**
> We thank to the reviewer for highlighting this important aspect. We acknowledge that there is a possible mismatch between benchmark outcomes utilizing in-sillico evaluators versus real world outcomes. However, we have tried to reduce the gap by carefully splitting the sequences between the “training” and “extrapolation” regions to ensure separation in both sequence and fitness spaces. The generative models and scorers have been trained on "training region" only and the in-sillico evaluator have been trained on whole dataset.
>
> ### **Q3: ablation studies for triplet construction**
> We thank the reviewer for their excellent suggestion. We have added additional experimentations regarding  how to create triplet specifically on 1) being generated from consecutive vs non-consecutive bins 2) whether using sequence similarity constrain between prompt and generated sequences. We have reported the results in section 6.1 in main text and section D in supplementary. The results highlight that naively creating preference data from non-consecutive pairs would confuse the model's learning and worsen the performance on 3 out of 4 splits. We hypothesize that, in order to properly learn from non-consecutive bins, one needs to develop a more sophisticated prompt to incorporate the distance between bins as well. Similarly, the results suggest that max mutation 15 between seed and desired/undesired sequences would slightly improve the performance on hard splits of AAV and GFP datasets while worsen it for medium splits. However, calculating sequence similarity for large dataset is computationally expensive therefore we would not suggest it as our default.
>
> ### **Q4: clarification question regarding on how to balance between exploration and exploitation**
> We thank the reviewer for their clarification question regarding balancing between exploration and exploitation. Temperature ($\tau$) is one of the main hyper-parameters that creates balance between exploration and exploitation. Temperature ($\tau$) hyper-parameter is used to control the randomness in generation and as a way to balance between greedy search (token with max probability when
> $\tau \rightarrow 0$) and uniform sampling (when $\tau \rightarrow \infty$). As highlighted from the results of the hyper-parameter tuning in section F in supplementary, in order to increase the diversity when scorer is used in inference, it is recommended to increase the temperature from 0.7 to 1.0.

---

> > ### Comment · Reviewer_jZ4t · 2024-11-26
> > **Post-rebuttal comment**
> >
> > Thank you for the effort in preparing rebuttal. Authors have mostly resolved my questions, and I'm keeping the score positive (6).

---

> > > ### Author Response · Authors · 2024-12-01
> > > **Thank you**
> > >
> > > We really appreciate the reviewer for helping us improving the manuscript quality.
> > >
> > > Kind Regards, Authors

---

### Official Review · Reviewer_P9DR · 2024-11-04

**Soundness:** 2
**Presentation:** 2
**Contribution:** 3
**Rating:** 8
**Confidence:** 3

**Summary:**

In order to optimize a certain protein of interest, some preliminary source of supervised data is collected around the property of interest and some mutational landscape of that protein. In the effort to achieve a higher functional capability than those derived from the supervised set, protein designers will propose, express, and assay new designs in hopes that they can achieve more fit proteins than their initial candidate pool. To that end, this work uses some of the reward modeling techniques from preference optimization literature to improve a conditional protein generation model’s ability to extrapolate to unseen fitness values. It argues to use the exact preference optimization (EXO) framework with a mixture of hard negatives to improve the generative model’s capability to sample fit sequences.

To assess the contribution of different components, it ablates the choice of preference optimization method and triplet generation techniques. In addition, it studies the performance of methods that explicitly use and do not use the scorer during sampling.

**Strengths:**

The model’s intervention is well motivated and deserves recognition for its simplicity. Preference optimization is easy to implement, and the results against non-scoring guided samplers make this an intervention that’s likely to be adopted. I can see this becoming a well adopted strategy for these reasons.

In introducing the intervention, the work does a good job in beginning to analyze what alterations are causal in increased performance. The experiments with respect to triplet curation and reward methods are well thought out and thorough. Each study is run with 5 replicates to ensure that performance is not a result of randomness. In addition, Figure 2 is quite compelling towards understanding EXO’s performance compared to other alternatives, while also illuminating the core goal of the project.

**Weaknesses:**

As the paper notes, all experiments are derived from a blending of wet-lab data and estimated wet-lab experimental outcomes. As such, there might be a mismatch between the benchmark outcomes and real world outcomes. This should not be counted against the work too much, but kept in the back of readers minds.

The two major areas for improvement in this work lie in elements of clarity and the derivation of baselines. Overall, the work asks the right questions, but the exposition in how the answers are derived leaves out critical details that makes it hard to have conviction in the whole outcomes.

On the note of clarity, the work finds certain key components unexplained. In particular, while dedicating an entire subsection to the role of sampling with a scorer, the details of how this performs are left unstated. Furthermore, the exact reasoning behind certain choices is left unsaid: why does sampling with a scorer have a different temperature and what is the maxim used to select this? As a minor nitpick, on line 284, the mentioning of $\beta=0.1$ should clarify a preference loss $\beta$ as it’s confusable with AdamW’s two separate $\beta$s.

My largest source of concern in the work centers around how baselines are evaluated and the steps leading to a hyperparameter selection. Looking at Table 11, hyperparameter choice can mean the difference between a SoTA performance  and unusable model. With this in mind, and the explanations towards those chosen for the preference models, the leading question then becomes how sensitive are the baseline models to hyperparameter selection? The text does not explain how they were optimized.

Even for certain choices in the paper, the statements: “[b]ased on the results shown in Figure 21 and Table 10, $\beta = 0.1$ is performing well” and “[s]imilarly, based on Figure 22 and Table 11, $\tau=0.7$ is performing well” led to confusion amongst what “performing well” means. Additionally, in what order were $\beta$ and $\tau$ chosen? Even if not a hard and fast rule it would aid the reader to understand which metrics were being balanced to make this decision.

Lastly, a more minor comment, the claims around Figure 3 should be hedged to take into account the stochastic nature of t-SNE and the relationships that they may imply. Both panels of the figure are challenging to read from the choice in symbols due to small size. It’s challenging to draw the same conclusion as the paper about the “supports by unseen ground truth extrapolation sequences.” This feedback seems more splitting hairs about a qualitative study so I don’t think it’s a strong critique, rather room for the wording to get a little qualification.

**Questions:**

* How is the sampling with scorer guidance performed?
* In what order were $\beta$ and $\tau$ determined?
* What choices went into the selection of hyperparameters for the baseline models?
* How does performance of the baseline models change when benchmarked with the same attention as the proposed model?

Because of the simplicity, this algorithm seems like an attractive solution for protein designers, but still has room to grow towards evaluation. To clarify, it does not need uniformly SoTA results to merit publication, rather reproducible experimental parameters to give confidence in performance differences. If work evolves to address the above questions, I’d happily raise my score.

---

> ### Author Response · Authors · 2024-11-25
> **Response to Reviewer P9DR**
>
> We appreciate the thoughtful comments and summary from reviewer P9DR, which have helped improve the quality of the manuscript.
>
> ### **W1: regarding the mismatch between the benchmark outcomes and real world outcomes**
> Thanks to the reviewer for highlighting this important aspect. We acknowledge that there is a possible mismatch between benchmark outcomes and real world outcomes. However, we have tried to reduce the gap by carefully splitting the sequences between the “training” and “extrapolation” regions to ensure separation in both sequence and fitness spaces.
>
> ### **W2: clarification regarding hyper-parameter tuning for proposed method**
> Thanks to the reviewer for highlighting this important clarification question. Based on the hyper-parameter tuning shown in section F in main text, we have chosen temperature 0.7 for non-scorer and 1.0 for when we are using scorer in inference. We have modified the section 5 in main text to clarify more regarding the hyper-parameter tuning for proposed method.
>
> ### **W3: clarification regarding hyper-parameter tuning for baselines**
> Thanks to the reviewer for highlighting this important concern. We have performed a thorough hyper-parameter tuning across all baselines and reported the results in section G. We have updated the results of the baseline methods according to their best hyper-parameters. Therefore, we have updated Figure 2 and Tables 2 and 5. The main conclusions after a thorough hyper-parameter tuning remain consistent with our previous findings. The main reason for the consistency is that reference papers [1] and [2] did thorough hyper-parameter tuning in their papers for the same datasets.
>
> ### **W4: suggestion regarding Figure 3**
> We thank the review for their constructive feedback. We have moved the figure 3 to supplementary and added more details regarding new experimentations (consecutive vs non-consecutive bins, sequence similarity constrains) and hyper-parameter tunings.
>
> ### **Q1: How is the sampling with scorer guidance performed?**
> We thank the reviewer for their clarification question. We have added more details in the inference section (section 3.6). Specifically, at iteration t, suppose there are M seeds available. We will generate N sequences per seed and choose the best one based on scorer as seed for next iteration.
>
> ### **Q2: In what order were $\beta$ and $\tau$ determined?**
>
> We thank the reviewer for their clarification question. We have tuned the $\beta$ first since it is the hyper-parameter used in fine-tuning and then we have tuned the temperature ($\tau$) since it is used in the inference. We have added these details in section 5 in main text.
>
> ### **Q3: What choices went into the selection of hyper-parameters for the baseline models?**
> We thank the reviewer for their clarification question. In the first submission, we didn’t perform hyper-parameter selection for baseline models since most of the state-of-the-art models ([1] and [2]) have been already reported on the same datasets/splits. Therefore, they have already done the hyper-parameter selection in their respective papers. We have utilized their default values reported in their papers or Github. However, in the revised manuscript, we performed hyper-parameter tuning as per reviewer's recommendation.
>
> ### **Q4: How does performance of the baseline models change when benchmarked with the same attention as the proposed model?**
>
> After thorough hyper-parameter tuning for all baselines methods with the same focus as given to the proposed methods, we can observe that the main conclusions remain consistent with our previous findings. We want to re-iterate that this finding is due to reference papers ([1] and [2]) having done their own thorough hyper-parameter tuning in their papers for the same datasets.
>
> [1] Kirjner, Andrew, et al. "Improving protein optimization with smoothed fitness landscapes. ICLR, 2024.
>
> [2] Lee, Minji, et al. “Robust Optimization in Protein Fitness Landscapes Using Reinforcement Learning in Latent Space” ICML 2024

---

> > ### Author Response · Authors · 2024-12-01
> > **Further discussion**
> >
> > We really appreciate the reviewer for helping us improving the manuscript quality. We would like to ask if our response has addressed your concerns? If not, we are very happy to discuss and further clarify.
> >
> > Kind Regards, Authors

---

> ### Comment · Reviewer_P9DR · 2024-12-01
> **Response to Authors**
>
> Each of my points were thoughtfully addressed in the revised version of the manuscript. The work now contains thorough descriptions of the methods, how baselines were tuned to create fair evaluations, and all relevant results were updated in line with this. I believe that future researchers will be able to build on the perspective of this work and trust the findings. I raise my score to reflect the growth of the work.
>
> Some minor aesthetic nitpicks: elements in Figure 1 aren't spaced correctly (e.g. the sub boxes in triplet curation aren't centered) and could benefit from a larger font for readability utilizing the negative space. Not a major contention, but would improve a camera ready version.

---

### Official Review · Reviewer_jBZU · 2024-11-05

**Soundness:** 3
**Presentation:** 3
**Contribution:** 3
**Rating:** 6
**Confidence:** 3

**Summary:**

In this work, preference learning on triplets of sequences has been proposed to improve the performance of prior edit-based methods for the task of protein design. In prior edit-based method(s), the conditional generative model is only trained on pairs of sequences where the likelihood is maximized for generating the sequence with higher property conditioned on the sequence with lower property. However, faced with three (or higher)-way relationships, the conditional generation is not guaranteed to move the optimization towards the sequence with the highest property. To mitigate this issue, the set(s) of sequence triplets are designed to further optimize the conditional generative model through preference learning losses.

Protein optimization tasks (four in total) with varying degree of extrapolation difficulty (designed by prior work) have been used to demonstrate the effectiveness of preference learning on sequential design of proteins. The proposed method works better than prior protein design approaches in terms of extrapolation and finding higher fitness samples in three out of the four design tasks.

**Strengths:**

1. Demonstrating the impact of preference learning in extrapolative generative protein design.

2. Nice ablation study on the impact of a) the strategies/criteria to create dataset of triplet sequences for preference learning, b) the type of preference learning approach adopted.

**Weaknesses:**

1. The authors have specified a separate category of methods for extrapolation, whereas by the definition of the paper in section 3.1, almost all the methods for protein optimization are doing extrapolation. They are all searching for sequences not existing in the train set that have properties beyond the range of training samples. The extent of extrapolation may differ from one work to another as discussed by  [1] and [2] that introduces the concept of “separation”, which can be seen as the extent of extrapolation, and the challenges associated with that in protein optimization tasks.

2. Given that this work is for protein optimization, the related work section should contain a category for methods that perform optimization in the latent space which could be guided by property values as well. The famous work of [3] is one of these methods which has been included and discussed in many protein optimization benchmarks. Also, the recent method of [2] (published at ICLR 2024), encodes the _preference_ of sample generation into the continuous latent space of the generative model. This makes the optimization robust to the extent of “separation", i.e., extrapolation, in the design tasks.

References.
1) Kirjner, Andrew, et al. "Improving protein optimization with smoothed fitness landscapes. ICLR, 2024.
2) Ghaffari, Saba, et al. "Robust Model-Based Optimization for Challenging Fitness Landscapes. ICLR, 2024.
3) Gómez-Bombarelli, Rafael et al. “Automatic Chemical Design Using a Data-Driven Continuous Representation of Molecules.”

**Questions:**

1. How were the pairs with true scores generated? Has the same binning of fitness been used to generate pairs as well as the triplets? Were the pairs taken from two consecutive bins?
In section 3.3 there is no mention of using the training pairs with their original scores, while it is mentioned in section 4.3 (first paragraph). Make the two consistent to avoid confusion.

2. Why are the triplets generated from consecutive bins? Is there an underlying assumption that the sequences coming from two adjacent bins are more similar than the ones further apart? The fitness landscape of protein does not necessarily behave like this. The fitness can have sharp variations in a local neighborhood of sequence space.

3. How is a small meaningful improvement of score determined for the perturbed seed sequence (section 3.3, line 163). I am assuming that having a small improvement is the requirement of the conditional modelling used. How is a meaningful improvement identified? Also, how different can a perturbed sequence look from the seed sequence? Is there any restriction on the dissimilarity between the perturbed and seed sequence in the sequence space?

4. In the main text you should clarify what $r_\phi$ (reward) stands for.

5. Please explain top-k and top-p sampling and the role of temperature in the inference.

6. Is the default operating mode of EXO with the “All” triplet creation (Table 2)?

7. (Minor). Change 2023 to 2024 in the citation for,
Improving protein optimization with smoothed fitness landscapes. In The Twelfth International Conference on Learning Representations, 2023.

---

> ### Author Response · Authors · 2024-11-25
> **Response to Reviewer jBZU Part 1**
>
> We sincerely appreciate reviewer jBZU for their thoughtful feedback, which has significantly improved the manuscript's quality.
>
> ###  **W1: Addressing the "separation" concept introduced by [2] versus "extrapolation" concept introduced by [1,4] and our paper**
>
> We thank the reviewer for highlighting this concern. We were not aware of [2] introducing the “separation” concept which is closely related to “extrapolation” region. As pointed out by the reviewer, the exact definition of “extrapolation” and “separation” are different. For example, we have considered “extrapolation” in both fitness and sequence space. Following [1], all sequences in the training data have mutational gap of 6 or 7 meaning they are 6 or 7 mutations away from top 1% sequences in the extrapolation region. In addition, there is a clear separation in the fitness landscape as well where the scorer and generative models are trained only on “training region” with noisy understanding of the “extrapolation” region.
>
> ### **W2: Addressing "optimization in latent space" in related work**
>
> Following reviewer’s recommendation, we have added a category in the related work section for including “protein optimization in the latent space”. We included the recommended references ([2] and [3]) as well as one of the-state-of-the-art models we have compared with [4].  The latter reference has utilized latent space of protein large language model (ESM-2 with 650M parameters). They have introduced a novel reinforcement learning approach to navigate through the latent space. Our benchmark has shown that our model has outperformed theirs in 3 out of 4 splits.
>
> ### **Q1: clarification regarding how pairs are generated**
>
> We thank the reviewer for their clarification question. We have followed the recommendation from [5] for pair creation. We generated perturbed sequences by masking-infilling starting (5% masking which is around 2 mutations for AAV with length of 28 and 14 mutations for GFP with the length of 280) from the training sequences (seeds). Scorer function is utilized to assess whether the newly generated pair (seed, sequence) has a small but meaningful improvement (e.g. 0.5) in the desired direction. We have not performed ablation studies to assess the impact of mask-infilling approach for pair creation versus pairs created from consecutive bins.
>
> ### **Q2: triplets generated from consecutive vs non-consecutive bins**
>
> We thank the reviewer for their insightful questions that have significantly helped improved the revised paper.  We have added an additional experimentation to addressing the consecutive vs non-consecutive bins. We have created preference data from non-consecutive bins in similar fashion as we have done in our default version (consecutive bins). We have reported the results in section 6.1 in main text and section D in supplementary. The results highlight that naively creating preference data from non-consecutive pairs would confuse the model's learning and worsen the performance on 3 out of 4 splits. We hypothesize that, in order to properly learn from non-consecutive bins, one needs to develop a more sophisticated prompt to incorporate the distance between bins as well. Due to the limited time we could not implement this but we think the model should benefit from it and can be considered as a future direction.
>
> ### **Q3: clarification question regarding "meaningful improvement" hyper-parameter in pair generation**
>
> We thank the reviewer for their clarification question. We have followed the recommendation from [5] to set the hyper-parameters (e.g. max fitness differences between pairs, or number of masking, etc) for pair creation and perturbations. We have added more details in the section 4.3 of main text.
>
> [1] Kirjner, Andrew, et al. "Improving protein optimization with smoothed fitness landscapes. ICLR, 2024.
>
> [2] Ghaffari, Saba, et al. "Robust Model-Based Optimization for Challenging Fitness Landscapes. ICLR, 2024.
>
> [3] Gómez-Bombarelli, Rafael et al. “Automatic Chemical Design Using a Data-Driven Continuous Representation of Molecules.”
>
> [4] Lee, Minji, et al. “Robust Optimization in Protein Fitness Landscapes Using Reinforcement Learning in Latent Space.” ICML 2024
>
> [5] Padmakumar, Vishakh, et al. "Extrapolative controlled sequence generation via iterative refinement." ICML 2023.

---

> > ### Comment · Reviewer_jBZU · 2024-12-01
> > **Response to the authors**
> >
> > Thanks for addressing my concerns.  I will raise my score to 6.
> > For sanity, the work by [2] looks at extrapolation in both sequence and fitness space.

---

> ### Author Response · Authors · 2024-11-25
> **Response to Reviewer jBZU Part 2**
>
> ### **Q4: sequence similarity constrain for pair and triplet generation**
>
> We thank the reviewer for their insightful question. We have not considered any sequence similarity constrain for pair creation. Since the focus of the paper was on triplet creation rather than pairs, we followed reviewer’s recommendation and did an ablation studies for sequence similarity in preference data creation. Results that are reported in section 6.1 of main text  and section D.5 of supplementary suggest that max mutation 15 between seed and desired/undesired sequences would slightly improve the performance on hard splits of AAV and GFP datasets while worsen it for medium splits. However, calculating sequence similarity for large dataset is computationally expensive therefore we would not suggest it as our default.
>
> ### **Q5: Clarification regarding what $r_{\phi}$ (reward) stands for.**
> We thank the reviewer for their clarification question. Due to the page limitation, we have added their exact explanations in the section A of supplementary.
>
> ### **Q6: Clarification regarding top-p, top-k and role of temperature**
> We thank the reviewer for their clarification questions. We added few lines in the inference (section 3.6), to explain top-p , top-k and temperature.
>
> ### **Q7: the default operating mode of EXO with the “All” triplet creation (Table 2)**
> We thank the reviewer for their clarification question. No it is EXO with the “Mistakes” triplet creation. We have added clarification in the paper (Table 3).
>
> ### **Q8: updating citation "Improving protein optimization with smoothed fitness landscapes."**
> We thank the reviewer for their attention to details. We have modified it in the revised paper.

---

> ### Author Response · Authors · 2024-12-01
> **Further discussion**
>
> We really appreciate the reviewer for helping us improving the manuscript quality. We would like to ask if our response has addressed your concerns? If not, we are very happy to discuss and further clarify.
>
> Kind Regards, Authors

---

### Author Response · Authors · 2024-11-25
**Global Rebuttal**

We sincerely thank the reviewers for their thoughtful and constructive feedback, which has greatly contributed to improving the quality and clarity of our manuscript. We summarize the major updates below and changes made in response to the reviewers’ comments:


**Summary of changes**

1. Additional experiments (Responding to reviewer **jBZU**):
    1. Preference data created from non-consecutive bins vs consecutive bins (default): Based on the result shown in Table 4, we observed that triplets created from non-consecutive bins worsen the performance of the proposed methods. We hypothesize that preference data created from non-consecutive bins confuses the the models by the different jumps that it should learn at the same time. The model might need an additional prompt to guide it through the jumps for non-consecutive bins unlike in the consecutive bin scenario. Due to the limited time we could not implement this but we think the model should benefit from it and can be considered as future work.
    2. Preference data created with sequence similarity constrain: Based on the result shown in Table 4, we have observed that max mutation 15 between seed and desired/undesired sequences would slightly improve the performance on hard splits of AAV and GFP datasets while worsen it for medium splits. However, calculating sequence similarity for large dataset is computationally expensive therefore we would not suggest it as our default.. We have explained in more details in section 6.1 of main text and section D.5 of supplementary.
2. Hyper-parameter tuning for baseline methods (Responding to reviewer **P9DR**):
    1. We have performed a thorough hyper-parameter tuning for all baseline methods and reported them in section G. We have updated the results of the baseline methods according to their best hyper-parameters. Therefore, we have updated Figure 2 and Tables 2 and 5.
3. Calculating computational cost of creating triplets based on default or scorer-distillation approach (Responding to reviewer **jZ4t**): As mentioned in the revised paper, scorer distillation is more than 200 times more expensive to run on GPU machines in comparison to the default for GFP dataset (280 length protein).
4. Added more explanation in inference section 3.6 (Responding to reviewer **jBZU**, **P9DR**):
    1. Added explanation for top-p, top-k and role of temperature in inference.
    2. Added explanation for inference with scorer
5. Add more explanation for scorer-distillation approach and its hardness definition in section 6.1 (Responding to reviewer **qbi3**).

**Final Remarks**

We hope these revisions and clarifications address the reviewers' concerns and further demonstrate the novelty and importance of our work. We remain grateful for the reviewers' insightful feedback, which has strengthened our manuscript significantly. Thank you for your time and consideration.

---

### Meta-Review · Area_Chair_A7mc · 2024-12-20

**Metareview:**

This paper proposes a preference learning approach on triplets of protein sequences to enhance protein design performance. By leveraging triplet-based preference learning, the method addresses the challenge of three-way relationships, improving the generative model's ability to produce sequences with higher properties compared to prior edit-based methods, which are only optimized on pairs of sequences. Evaluations on GFP and AAV datasets show that the proposed method outperforms state-of-the-art scorer-based and non-scorer-based baselines in terms of extrapolation rate and top-100 fitness.

Generally, the reviewers find the contribution of introducing triplet-based preference learning to protein design both novel and valuable. The approach is well-motivated, with a clear rationale for using preference optimization to generate protein sequences with fitness values beyond those observed in training. The method is simple to implement and shows promising results, particularly when compared to non-scoring guided samplers, making it a promising candidate for widespread adoption. The original manuscript lacks clarity on key components, such as scorer-based sampling and hyperparameter selection, and is missing a more comprehensive discussion of latent-space optimization methods, broader dataset evaluations, real-world validation, and the addressing of scalability and robustness to noisy labels. The authors’ responses together with additional experimental results have successfully addressed the reviewers’ concerns.

All reviewers recommend acceptance. I support the reviewers’ recommendation. Authors should attend to main points in the reviews, especially additional analyses and experiments, when preparing a final version.

**Additional Comments On Reviewer Discussion:**

The authors’ responses together with additional experimental results have successfully addressed the reviewers’ concerns. All reviewers recommend acceptance. I support the reviewers’ recommendation. Authors should attend to main points in the reviews, especially additional analyses and experiments, when preparing a final version.

---

### Decision · Program_Chairs · 2025-01-22

Accept (Poster)